# A Simple yet Powerful Deep Active Learning with Snapshots Ensembles

**Seohyeon Jung**\* **Sanghyun Kim**\* **Juho Lee**
Kim Jaechul Graduate School of AI
Korea Advanced Institute of Science and Technology (KAIST)
Daejeon, Republic of Korea
{heon2203,nannullna,juholee}@kaist.ac.kr

## Abstract

Given an unlabeled pool of data and the experts who can label them, active learning aims to build an agent that can effectively acquire data to be queried to the experts, maximizing the gain in performance when trained with them. While there are several principles for active learning, a prevailing approach is to estimate uncertainties of predictions for unlabeled samples and use them to define acquisition functions. Active learning with the uncertainty principle works well for deep learning, especially for large-scale image classification tasks with deep neural networks. Still, it is often overlooked how the uncertainty of predictions is estimated, despite the common findings on the difficulty of accurately estimating uncertainties of deep neural networks. In this paper, we highlight the effectiveness of snapshot ensembles for deep active learning. Compared to the previous approaches based on Monte-Carlo dropout or deep ensembles, we show that a simple acquisition strategy based on uncertainties estimated from parameter snapshots gathered from a single optimization path significantly improves the quality of the acquired samples. Based on this observation, we further propose an efficient active learning algorithm that maintains a single learning trajectory throughout the entire active learning episodes, unlike the existing algorithms training models from scratch for every active learning episode. Through the extensive empirical comparison, we demonstrate the effectiveness of snapshot ensembles for deep active learning. Our code is available at: https://github.com/nannullna/snapshot-al.

## 1 Introduction

The progress of deep learning is largely driven by data, and we often work with well-curated and labeled benchmark data for model developments. However, in practice, such nicely labeled data are rarely available. Many of the data accessible to practitioners are unlabeled, and more importantly, labeling such data incurs costs due to human resources involved in the labeling process. Active Learning (AL) may reduce the gap between the ideal and real-world scenarios by selecting the informative samples from the unlabeled pool of data, so after being labeled and trained with them, a model can maximally improve the performance.

The main ingredient of an AL algorithm is the acquisition function which ranks the samples in an unlabeled pool with respect to their utility for improvement. While there are several possible design principles (Ren et al., 2021), in this paper, we mainly focus on the acquisition functions based on the uncertainty of the predictions. Intuitively speaking, given a model trained with the data acquired so far, an unlabeled example exhibiting high predictive uncertainty with respect to the model would be a "confusing" sample which would substantially improve the model if being trained with the label acquired from experts. A popular approach in this line is Bayesian Active Learning by Disagreement (BALD) (Houlsby et al., 2011), where a committee of multiple models predicts an unlabeled sample, and the degree of disagreement is measured as a ranking factor. Here, the multiple models are

---

\* Equal contribution

usually constructed in a Bayesian fashion, and their disagreement reflects the model uncertainty about the prediction. BALD is demonstrated to scale well for modern deep neural networks for high-dimensional and large-scale data (Gal et al., 2017).

Similar to BALD, many AL algorithms based on uncertainty employ a committee of models to estimate the uncertainty of predictions. The problem is, for deep neural networks trained with high-dimensional data, it is often frustratingly difficult to accurately estimate the uncertainty. To address this, Gal et al. (2017) proposed to use Monte-Carlo DropOut (MCDO) (Gal and Ghahramani, 2017), an instance of variational approximation to the posteriors and predictive uncertainty, while Rakesh and Jain (2021) suggested using more generic spike-and-slab variational posteriors (Louizos et al., 2017). Nevertheless, variational approximations tend to underestimate the posterior variances (Blei et al., 2017; Le Folgoc et al., 2021), so the uncertainty-based acquisition functions computed from them may be suboptimal. Alternatively, one can employ Deep Ensemble (DE) (Lakshminarayanan et al., 2017), where a single model is trained multiple times with the same data but with different random seeds for initialization and mini-batching. Despite being simple to implement, DE works surprisingly well, surpassing most of the Bayesian Neural Networks (BNN) alternatives in terms of accuracy and predictive uncertainty (Fort et al., 2021; Ovadia et al., 2019). To this end, Beluch et al. (2018) highlighted the effectiveness of DE as a way to estimate uncertainty for acquisition functions and demonstrated excellent performance.

A drawback of DE is that it is computationally expensive, as it requires multiple models to be trained and maintained for inference. As an alternative, Snapshot Ensemble (SE) (Huang et al., 2017; Garipov et al., 2018) proposes to collect multiple model snapshots (checkpoints) within a single learning trajectory, rather than collecting them at the end of the multiple learning trajectories as in DE. Compared to DE, SE enables the construction of a decent set of models without having to go through multiple training runs while not losing too much accuracy.

Inspired by the advantage of SE, we study its use in the context of AL. Specifically, we estimate the uncertainties from SE and use them to evaluate the uncertainty-based acquisition functions. Through extensive empirical comparisons, we demonstrate that the AL based on SE significantly outperforms existing approaches, even comparable to or better than the one with DE. This result is somewhat surprising since it is often reported that SE is less accurate than DE (Ashukha et al., 2020). Moreover, based on this observation, we propose a novel AL algorithm that can substantially reduce the number of training steps required until the final acquisition. Typically, an AL algorithm alternates between acquiring labels based on a model and re-training the model with the newly acquired labels. Here, for every re-training step, the old models are discarded, and a new model is trained from scratch. Instead, we suggest maintaining a model on a single learning trajectory throughout the entire AL procedure and gathering snapshots from the trajectory to compute acquisition functions. We show that this can significantly reduce the number of training steps without sacrificing too much accuracy. In summary, our contributions are as follows:

- We propose to use SE for the uncertainty-based acquisition functions for AL and demonstrate its effectiveness through various empirical evaluations.
- We propose a novel AL algorithm where a single learning trajectory is maintained and used to compute acquisition functions throughout the entire AL procedure. We demonstrate that our algorithm could achieve decent accuracy with much fewer training steps.

## 2 BACKGROUND

### 2.1 SETTINGS AND BASIC ACTIVE LEARNING ALGORITHM

In this paper, we mainly discuss $K$-way classification problem, where the goal is to learn a classifier $f(\cdot; \boldsymbol{\theta})$, parameterized by $\boldsymbol{\theta}$, taking an input $\boldsymbol{x} \in \mathbb{R}^d$ to produce a $K$-dimensional probability vector, that is, $f(\boldsymbol{x}; \boldsymbol{\theta}) \in [0,1]^K$ such that $\sum_{k=1}^K f_k(\boldsymbol{x}; \boldsymbol{\theta}) = 1$. To learn $\boldsymbol{\theta}$, we need a labeled dataset consisting of pairs of an input $\boldsymbol{x}$ and corresponding label $y \in \{1, \ldots, K\}$, but in AL, we are given only an unlabeled dataset $\mathcal{U} = \{\boldsymbol{x}_i\}_{i=1}^n$ without labels.

An AL algorithm is defined with a classifier model $f(\cdot; \boldsymbol{\theta})$ and an *acquisition function* $a : \mathbb{R}^d \to \mathbb{R}$ measuring how useful an unlabeled example $\boldsymbol{x}$ is to the classifier $f(\cdot; \boldsymbol{\theta})$. Given $f$ and $a$, an AL algorithm alternates between acquiring the labels for chosen unlabeled samples and training the

classifier with the labeled samples. A single iteration of acquiring samples and training the classifier is called an *episode*. In the first episode, $m$ samples are randomly chosen from $\mathcal{U}$, and the labels are acquired for them to constitute an initial training set $\mathcal{D}_{\text{train}}$. The classifier is then trained with $\mathcal{D}_{\text{train}}$ to obtain $\boldsymbol{\theta}_1$, or in case of the ensemble-based AL, a set of parameters $\{\boldsymbol{\theta}_1^{(s)}\}_{s=1}^S$. The labeled samples are removed from $\mathcal{U}$. For all subsequent episodes $t \geq 2$, with the parameters $\{\boldsymbol{\theta}_{t-1}^{(s)}\}_{s=1}^S$ from the previous episode, the samples remaining in $\mathcal{U}$ are ranked with the values of the acquisition function, and the top $m$ of them are selected to be labeled. The newly labeled $m$ samples are then appended to the labeled training set $\mathcal{D}_{\text{train}}$, and the classifier is trained *from scratch* with the extended $\mathcal{D}_{\text{train}}$ to obtain $\{\boldsymbol{\theta}_t^{(s)}\}_{s=1}^S$. The algorithm terminates when it reaches the predetermined number of episodes $T$, and the goal of AL is to maximize the accuracy of the classifier after the final episode.

## 2.2 ACQUISITION FUNCTIONS

Here, we review some popular choices for the acquisition functions, especially the ones based on predictive uncertainties. We consider the following acquisition functions which estimate the uncertainty of the predictions via a set of parameters $\{\boldsymbol{\theta}^{(s)}\}_{s=1}^S$. This set of parameters defines a committee of models $\{f(\cdot; \boldsymbol{\theta}^{(s)})\}_{s=1}^S$.

**Maximum Entropy (ME).** ME measures the predictive entropy of a given example $\boldsymbol{x}$ which can be approximated as,

$$H[\boldsymbol{y}|\boldsymbol{x}, \mathcal{D}_{\text{train}}] = -\sum_{k=1}^K p(y = k|\boldsymbol{x}, \mathcal{D}_{\text{train}}) \log p(y = k|\boldsymbol{x}, \mathcal{D}_{\text{train}})$$

$$\approx -\sum_{k=1}^K \left(\frac{1}{S}\sum_{s=1}^S f_k(\boldsymbol{x}; \boldsymbol{\theta}^{(s)})\right) \log \left(\frac{1}{S}\sum_{s=1}^S f_k(\boldsymbol{x}; \boldsymbol{\theta}^{(s)})\right). \tag{1}$$

Larger entropy means that the model is more uncertain about the prediction.

**Variation Ratio (VR).** VR (Freeman, 1965) measures how certain the model is about its prediction for $\boldsymbol{x}$, or how many of the committee agree with the prediction, and is calculated as $1 - f_m/S$, where $f_m$ is the frequency of a mode prediction over $S$ committee members. Similarly, Least Confident (LC) sampling chooses the least confident sample as $1 - \max_k p(y = k|\boldsymbol{x}, \mathcal{D}_{\text{train}})$, and Margin (MAR) sampling chooses examples with the smallest difference between the largest and the second largest probabilities.

**Bayesian Active Learning by Disagreement (BALD).** BALD (Houlsby et al., 2011) measures the mutual information between the label and the parameter given an input $\boldsymbol{x}$ and the training data $\mathcal{D}_{\text{train}}$. It can also be interpreted as measuring disagreement among the predictions of the committee members. BALD is maximized when each of the committee members is certain about their own predictions (small $H[y|\boldsymbol{x}, \boldsymbol{\theta}^{(s)}]$), but the predictions disagree with each other, so the averaged prediction becomes uncertain (high $H[y|\boldsymbol{x}, \mathcal{D}_{\text{train}}]$).

$$I[y, \boldsymbol{\theta}|\boldsymbol{x}, \mathcal{D}_{\text{train}}] = H[y|\boldsymbol{x}, \mathcal{D}_{\text{train}}] - \mathbb{E}_{\boldsymbol{\theta}|\mathcal{D}_{\text{train}}}\big[H[y|\boldsymbol{x}, \boldsymbol{\theta}]\big]$$

$$\approx -\sum_{k=1}^K \left(\frac{1}{S}\sum_{s=1}^S f_k(\boldsymbol{x}; \boldsymbol{\theta}^{(s)})\right) \log \left(\frac{1}{S}\sum_{s=1}^S f_k(\boldsymbol{x}; \boldsymbol{\theta}^{(s)})\right)$$

$$+ \frac{1}{S}\sum_{s=1}^S \sum_{k=1}^K f_k(\boldsymbol{x}; \boldsymbol{\theta}^{(s)}) \log f_k(\boldsymbol{x}; \boldsymbol{\theta}^{(s)}). \tag{2}$$

## 2.3 ESTIMATING UNCERTAINTY

In Bayesian AL, the model parameter $\boldsymbol{\theta}$ is treated as a random variable with prior $p(\boldsymbol{\theta})$ and the posterior $p(\boldsymbol{\theta}|\mathcal{D}_{\text{train}})$ is approximated via a set of samples $\{\boldsymbol{\theta}^{(s)}\}_{s=1}^S$. There are several ways to approximate it.

**Variational approximations.** For variational approximation, an easy-to-handle variational distribution $q(\boldsymbol{\theta})$ is introduced, learned to minimize $D_{\mathrm{KL}}[q(\boldsymbol{\theta})\|p(\boldsymbol{\theta}|\mathcal{D}_{\mathrm{train}})]$, and used as a proxy for $p(\boldsymbol{\theta}|\mathcal{D}_{\mathrm{train}})$. That is, once obtained approximate distribution $q(\boldsymbol{\theta})$, we draw $\boldsymbol{\theta}^{(1)}, \ldots, \boldsymbol{\theta}^{(S)} \overset{\mathrm{i.i.d.}}{\sim} q(\boldsymbol{\theta})$. A popular choice for $q(\boldsymbol{\theta})$ is mean-field Gaussian distribution (Blundell et al., 2015). In AL literature, MCDO is widely used after (Gal et al., 2017), where the dropout (Srivastava et al., 2014) is applied to the model $f$ and the randomness due to the dropout is interpreted as an approximate posterior $q(\boldsymbol{\theta})$. While relatively simple to implement, the variational approximations are known to underestimate the posterior variances.

**Deep ensembles.** DE (Lakshminarayanan et al., 2017) trains $f(\cdot; \boldsymbol{\theta})$ multiple times with the same $\mathcal{D}_{\mathrm{train}}$ but with different random initializations to obtain $\{\boldsymbol{\theta}^{(s)}\}_{s=1}^{S}$. DE is simple to implement, and yet its performance is remarkable, achieving state-of-the-art across various applications. The power of DE is mainly from its property to pick parameters from multiple modes (Fort et al., 2021), so the committee constructed from them yields a diverse set of predictions. Even if it is not explicitly assuming the prior for $p(\boldsymbol{\theta})$, DE can roughly be interpreted as an approximate Bayesian inference method (Wilson and Izmailov, 2021; D'Angelo and Fortuin, 2021), so the parameters $\{\boldsymbol{\theta}^{(s)}\}_{s=1}^{S}$ can be interpreted as posterior samples approximating the uncertainty of the models.

**Snapshot ensembles.** DE is expensive, both for training and inference since it has to keep multiple models with different parameters. SE (Huang et al., 2017; Garipov et al., 2018) is an idea to reduce the training cost of DE, where the multiple parameters $\{\boldsymbol{\theta}^{(s)}\}_{s=1}^{S}$ are gathered within a *single* training run rather than collected from multiple training runs. To obtain diverse parameters, the learning rate of the training run is *carefully* chosen to encourage the optimization path to explore a wide area of the loss surface, and the parameter "snapshots" are periodically captured during the training run. The SE usually underperforms DE with the same number of parameters gathered, but it can collect the parameters much faster since it requires only a single training run.

## 3 METHODS

### 3.1 ACTIVE LEARNING WITH SNAPSHOT ENSEMBLES

We first present an AL algorithm based on SE that is simple and efficient. In each episode, we store parameter snapshots at regular intervals during the classifier training stage, which are then used to compute the acquisition function at the end of the episode. This approach incurs *no additional computation cost for training*, unlike AL based on DE. In the final episode, we have several options to choose from when we train the classifier using the acquired data $\mathcal{D}_{\mathrm{train}}$: following a single learning trajectory and picking the parameter at the last step as a point estimate, or applying DE to obtain an ensembled model. Algorithm 1 summarizes our SE-based AL algorithm, where the final classifier is obtained with vanilla Stochastic Gradient Descent (SGD), but DE can be applied instead. In Section 5, we demonstrate that this simple modification with SE, albeit no increase in the training and inference costs, significantly improves the performance, even outperforming the AL with DE.

### 3.2 ACTIVE LEARNING WITH SNAPSHOT ENSEMBLES AND FINE-TUNING

Algorithm 1 trains classifiers for intermediate episodes and discards them after computing the acquisition function. However, considering that the acquisition function computed from a single learning trajectory works well, we can improve efficiency by keeping a single trajectory *throughout all episodes* and using the resulting parameter of the previous episode as *initialization for the next episode*. We call this strategy SE + FT. There are two things to note here. First, since we continuously fine-tune a single model, we need fewer training steps for each episode than the vanilla AL. Second, although the intermediate classifiers may be less accurate than those trained from scratch, as expected, this is less important since *what really matters is the accuracy after the final episode*. We argue that the intermediate episodes are to acquire samples quickly for the final training, so the accuracy of the classifier during that process is not important.

Also, our AL procedure with fine-tuning is reminiscent of continual learning, in the sense that a single model is continually being trained for multiple episodes having different data. To this end, we employ two commonly used tricks in continual learning literature.

**Algorithm 1:** AL with SE

**Input :** Unlabeled dataset $\mathcal{U}$, number of episodes $T$, number of (acquisitions $m$, snapshots $S$, SGD steps $N$) per episode, acquisition function $a$, snapshot threshold steps $N_{\text{thres}}$, objective function $\mathcal{J}$, learning rate schedule $\eta$.
**Ouput:** A classifier $f(\cdot; \boldsymbol{\theta}_*)$.
Randomly draw $m$ samples from $\mathcal{U}$, remove them from $\mathcal{U}$, and set them as $\mathcal{D}_{\text{train}}$.
**for** $t = 1, \ldots, T$ **do**
   Randomly initialize $\boldsymbol{\theta}_0$.
   Set $\boldsymbol{\Theta} \leftarrow \varnothing$.
   **for** $j = 1, \ldots, N$ **do**
      Draw a mini-batch $\mathcal{B}$ from $\mathcal{D}_{\text{train}}$.
      $\boldsymbol{\theta}_j \leftarrow \boldsymbol{\theta}_{j-1} - \eta(j) \nabla_{\boldsymbol{\theta}} \mathcal{J}(\mathcal{B}, \boldsymbol{\theta}_{j-1})$.
      **if** $j \geq N_{\text{thres}} \wedge$
        $\text{mod}\left(j - N_{\text{thres}}, \lfloor \frac{N - N_{\text{thres}}}{S} \rfloor\right) = 0$ **then**
        | $\boldsymbol{\Theta} \leftarrow \boldsymbol{\Theta} \cup \{\boldsymbol{\theta}_j\}$.
      **end**
   **end**
   **if** $t < T$ **then**
      Compute $a(\boldsymbol{x}, \boldsymbol{\Theta})$ for all $\boldsymbol{x} \in \mathcal{U}$.
      Pick top $m$ samples, remove them from $\mathcal{U}$, and append them to $\mathcal{D}_{\text{train}}$.
   **else**
      | Set $\boldsymbol{\theta}_* \leftarrow \boldsymbol{\theta}_N$.
   **end**
**end**

---

**Algorithm 2:** AL with SE + Fine-tuning (FT)

**Input :** Input for Algorithm 1 + regularization parameter $\lambda$.
**Ouput:** A classifier $f(\cdot; \boldsymbol{\theta}_*)$.
Randomly draw $m$ samples from $\mathcal{U}$, remove them from $\mathcal{U}$, and set them as $\mathcal{D}_{\text{train}}$.
Randomly initialize $\boldsymbol{\theta}_0$.
**for** $t = 1, \ldots, T$ **do**
   // Only at the final episode!
   **if** $t = T$ **then**
      | Randomly initialize $\boldsymbol{\theta}_0$.
   **end**
   Set $\boldsymbol{\Theta} \leftarrow \varnothing$.
   **for** $j = 1, \ldots, N$ **do**
      Draw a mini-batch $\mathcal{B}$ from $\mathcal{D}_{\text{train}}$.
      $\boldsymbol{\theta}_j \leftarrow \boldsymbol{\theta}_{j-1} -$
      $\eta(j)\nabla_{\boldsymbol{\theta}}(\mathcal{J}(\mathcal{B}, \boldsymbol{\theta}_{j-1}) + \lambda \mathbb{1}_{\{t>1\}} \|\boldsymbol{\theta}_{j-1} - \boldsymbol{\theta}_0\|^2)$.
      **if** $j \geq N_{\text{thres}} \wedge$
        $\text{mod}\left(j - N_{\text{thres}}, \lfloor \frac{N - N_{\text{thres}}}{S} \rfloor\right) = 0$ **then**
        | $\boldsymbol{\Theta} \leftarrow \boldsymbol{\Theta} \cup \{\boldsymbol{\theta}_j\}$.
      **end**
   **end**
   **if** $t < T$ **then**
      Compute $a(\boldsymbol{x}, \boldsymbol{\Theta})$ for all $\boldsymbol{x} \in \mathcal{U}$.
      Pick top $m$ samples, remove them from $\mathcal{U}$, and append them to $\mathcal{D}_{\text{train}}$
      // Reuse in the next episode.
      $\boldsymbol{\theta}_0 \leftarrow \boldsymbol{\theta}_N$.
   **else**
      | Set $\boldsymbol{\theta}_* \leftarrow \boldsymbol{\theta}_N$.
   **end**
**end**

---

**Replay buffer.** In principle, for the purpose of fine-tuning, we may use only the newly acquired data from the previous episode for fine-tuning. However, this would cause catastrophic forgetting (McCloskey and Cohen, 1989), so the acquisition function based on it may be biased towards recently acquired data. In order to prevent this, we adopt the idea of using a replay buffer (Jung et al., 2016; Rolnick et al., 2019; Aljundi et al., 2019), where we draw some portion of the data from the newly acquired data and the remaining portion from the past data. We empirically find that this significantly improves the stability of the acquisition functions.

**Regularization.** Similar to the replay buffer, we regularize the fine-tuning procedure to avoid deviating too much from the previous parameter (Kirkpatrick et al., 2017). That is, optimize the parameter $\boldsymbol{\theta}$ with the $\ell_2$-regularizer $\|\boldsymbol{\theta} - \boldsymbol{\theta}_0\|^2$ where $\boldsymbol{\theta}_0$ is the starting point of the fine-tuning (the parameters passed from the previous episodes). We also find that this regularization improves the quality of the acquisition functions, leading to better classification accuracy in the final episode.

Algorithm 2 summarizes the AL with fine-tuning. The parts that are different from the AL without fine-tuning are marked as blue.

## 4 RELATED WORKS

**Active Learning.** Based on how an unlabeled example is fed into an AL agent, AL can broadly be categorized into membership query synthesis, where the agent even generates examples in the sample space, stream-based selective sampling, where the agent decides whether a given input is helpful if labeled in online settings, and pool-based AL, where the agent can access a large unlabeled pool (Settles, 2009). Pool-based AL can be further divided into uncertainty-based approach, diversity-based approach, and hybrid approach (Ren et al., 2021). Geifman and El-Yaniv (2019) first introduced neural architecture search into AL, claiming that the over-parameterized model could lead

to overfitting especially in the earlier episodes, and therefore, uncertainty estimates could be inaccurate. Similarly, Munjal et al. (2022) argued that the optimal hyperparameters may vary with the size of labeled examples and used Bayesian hyperparameter optimization (AutoML) every episode.

**Ensemble.** Ensembles of neural networks have shown competitive performance improvement and are widely used in machine learning and deep learning. In addition, it shows improvements in estimates for predictive uncertainty. DE (Lakshminarayanan et al., 2017) is one of the well-behaved methods estimating the uncertainty of deep neural networks. This method works with several classifiers trained with the same dataset and architecture but with different seeds for a random number generator. However, the biggest limitation of ensemble is its computational costs because it requires training multiple models (Izmailov et al., 2018).

**Active Learning with Ensemble.** The uncertainty of an example for neural networks can be estimated by using an ensemble of neural networks in the context of AL. Gal et al. (2017) used BALD (Houlsby et al., 2011) on MCDO (Gal and Ghahramani, 2017) and later extended it to batch settings where it considers overlaps among data points to be acquired (Kirsch et al., 2019). However, due to the expensive cost of training multiple models, most research on ensemble-based AL had been restricted to the traditional ML algorithms (Melville and Mooney, 2004; Körner and Wrobel, 2006). Beluch et al. (2018) compares the performance of various acquisition functions and uncertainty estimation methods on large-scale image classification tasks, proving that DE consistently outperforms other uncertainty-based methods such as MCDO(Gal et al., 2017) or a single model. In the same manner, Bayesian neural networks show much robustness and reliability compared to DE and MCDO in the context of AL with continual learning (Rakesh and Jain, 2021).

## 5 EXPERIMENTS

In this section, through an extensive empirical comparison on three image classification benchmarks (CIFAR-10, CIFAR-100 (Krizhevsky et al., 2009), and Tiny ImageNet (Le and Yang, 2015)), we would like to demonstrate the followings:

- Measuring uncertainty via SE for AL is effective, comparable, or even better than the one based on DE across various choices of uncertainty-based acquisition functions.

- AL with SE + FT can significantly reduce the training cost without sacrificing too much accuracy.

- The reason why SE is effective for AL is that it can build a committee of models yielding diverse predictions. Interestingly, for the purpose of acquiring samples of better qualities, the diversity in predictions of the models is more important than the accuracy of the models.

We compare the AL algorithms with three variables – acquisition functions (VR, ME, BALD, MAR), algorithms to measure uncertainty (DE, SE, MCDO), and how to train the classifier in the final episode (single model via a vanilla SGD, DE). We report the results with ResNet-18 (He et al., 2016). Please refer to Appendix B for more details, such as experimental protocols or hyperparameter settings. The test accuracy results on CIFAR-10, CIFAR-100, and Tiny ImageNet are summarized in Table 1, Table 2, and Table 3, respectively, according to the proportion of labeled examples. Due to the limited resources, we report the results with four and three acquisition functions for CIFAR-10 and CIFAR-100, respectively, and the results with only VR for Tiny ImageNet.

### 5.1 ANALYSIS OF THE MAIN RESULTS

**Effectiveness of SE for AL.** Table 1 confirms that the measuring uncertainty with SE acquires the samples leading to the best classification accuracies, regardless of the choice of acquisition functions or how we train the final classifier. This is somewhat remarkable, considering that the runtimes of SE-based methods are not significantly longer than those for the random baselines, while DE requires significantly longer runtimes due to its requirement to train and test with multiple models. It is also noteworthy that the choice of uncertainty estimation method was much more crucial than the choice of acquisition functions. A similar trend is evident for CIFAR-100 and Tiny ImageNet, where SE generally produces the best classification accuracies. We set $S = 5$ for SE and DE and $S = 25$ for

**Table 1:** Test accuracy on CIFAR-10 according to the ratio of labeled examples

| Acq Fn | Uncertainty | A single model at the final episode | | | | DE at the final episode | | | | Runtime |
|---|---|---|---|---|---|---|---|---|---|---|
| | | 10% | 15% | 20% | 25% | 10% | 15% | 20% | 25% | |
| VR | SE | 69.10±0.46 | **79.60**±0.55 | 84.22±0.49 | **87.39**±0.26 | **75.01** | **83.38** | **87.22** | **89.75** | 15.3 hr |
| | SE + FT | **69.72**±0.57 | 79.36±0.33 | **84.24**±0.23 | 87.06±0.28 | 74.94 | 83.11 | 87.14 | 89.26 | 2.3 hr |
| | DE | 66.43±0.75 | 77.44±0.73 | 82.48±0.83 | 86.34±0.44 | 73.23 | 81.77 | 85.96 | 88.95 | 60.5 hr |
| | MCDO | 67.73±0.76 | 76.98±0.45 | 81.87±0.52 | 86.05±0.20 | 73.12 | 81.52 | 85.70 | 89.10 | 13.3 hr |
| BALD | SE | **70.80**±0.06 | **79.63**±0.37 | **84.32**±0.35 | **87.43**±0.27 | **76.37** | **83.19** | **86.86** | **89.25** | 15.3 hr |
| | SE + FT | 70.28±0.53 | 78.00±0.28 | 83.36±0.24 | 86.64±0.26 | 75.27 | 81.57 | 85.99 | 88.85 | 2.3 hr |
| | DE | 68.36±1.16 | 78.10±0.56 | 82.43±0.66 | 86.12±0.30 | 74.07 | 82.23 | 85.67 | 88.81 | 60.5 hr |
| | MCDO | 69.28±0.48 | 77.32±0.24 | 82.85±0.27 | 86.15±0.28 | 74.03 | 81.17 | 86.41 | 88.52 | 13.3 hr |
| ME | SE | 68.39±0.84 | 79.11±0.58 | 84.19±0.20 | 87.37±0.26 | 74.79 | 82.99 | 86.82 | 89.65 | 15.3 hr |
| | SE + FT | **70.25**±0.81 | **79.68**±0.18 | **84.56**±0.19 | **87.52**±0.19 | **76.00** | **83.47** | **87.33** | **89.69** | 2.3 hr |
| | DE | 65.77±1.39 | 77.31±0.85 | 82.54±0.26 | 86.24±0.29 | 72.30 | 82.08 | 85.97 | 88.97 | 60.5 hr |
| | MCDO | 67.96±0.89 | 77.67±0.67 | 82.70±0.68 | 86.79±0.28 | 74.43 | 81.89 | 85.94 | 89.31 | 13.3 hr |
| MAR | SE | 70.89±0.32 | 79.12±0.57 | 84.03±0.27 | 87.49±0.20 | 76.34 | 82.80 | 86.82 | 89.53 | 15.3 hr |
| | SE + FT | **73.26**±0.46 | **81.09**±0.27 | **85.43**±0.17 | **87.87**±0.17 | **77.95** | **83.93** | **87.88** | **89.70** | 2.3 hr |
| | DE | 70.48±0.59 | 79.79±0.47 | 83.48±0.29 | 87.03±0.39 | 76.49 | 83.47 | 86.78 | 89.51 | 60.5 hr |
| | MCDO | 71.82±0.32 | 79.25±0.44 | 82.75±0.39 | 85.19±0.19 | 75.77 | 82.15 | 84.98 | 87.32 | 13.3 hr |
| Random | | 69.88±0.25 | 78.08±0.37 | 82.44±0.38 | 84.58±0.09 | 74.85 | 81.32 | 84.95 | 86.92 | 12.1 hr |

**Table 2:** Test accuracy on CIFAR-100 according to the ratio of labeled examples

| Acq Fn | Uncertainty | A single model at the final episode | | | | DE at the final episode | | | | Runtime |
|---|---|---|---|---|---|---|---|---|---|---|
| | | 18% | 22% | 26% | 30% | 18% | 22% | 26% | 30% | |
| VR | SE | 58.58±0.19 | **62.75**±0.40 | **65.77**±0.27 | **68.02**±0.24 | 64.02 | **68.27** | 70.87 | **73.37** | 4.7 hr |
| | SE + FT | **59.04**±0.27 | 62.35±0.15 | 65.21±0.15 | 67.27±0.29 | **64.06** | 67.15 | 69.53 | 71.71 | 1.1 hr |
| | DE | 57.22±0.39 | 61.73±0.16 | 64.79±0.31 | 67.41±0.26 | 63.35 | 67.65 | 70.55 | 72.71 | 18.5 hr |
| | MCDO | 58.30±0.25 | 62.49±0.25 | 65.00±0.17 | 67.70±0.15 | 63.95 | 68.18 | 70.39 | 72.91 | 4.5 hr |
| BALD | SE | 57.13±0.52 | 61.10±0.42 | 63.85±0.39 | 66.05±0.36 | 61.86 | 65.51 | 68.39 | 70.63 | 4.7 hr |
| | SE + FT | 57.82±0.36 | 61.46±0.28 | 63.45±0.11 | 65.29±0.26 | 62.55 | 66.21 | 68.30 | 69.96 | 1.1 hr |
| | DE | **58.25**±0.42 | 62.26±0.32 | **65.40**±0.21 | **67.40**±0.21 | **63.28** | 67.04 | **70.27** | **72.18** | 18.5 hr |
| | MCDO | 58.03±0.29 | **62.28**±0.27 | 65.06±0.24 | 67.23±0.39 | 62.84 | **67.07** | 69.66 | 71.81 | 4.5 hr |
| ME | SE | 57.21±0.28 | 61.84±0.38 | 64.61±0.40 | **67.08**±0.25 | 62.73 | **67.73** | 70.36 | 72.24 | 4.7 hr |
| | SE + FT | **58.92**±0.25 | **62.41**±0.26 | **65.04**±0.22 | **67.08**±0.33 | **64.11** | 67.53 | 69.91 | 71.43 | 1.1 hr |
| | DE | 56.40±0.44 | 61.37±0.48 | 63.95±0.32 | 67.03±0.45 | 62.77 | 67.13 | 69.83 | **72.65** | 18.5 hr |
| | MCDO | 56.90±0.38 | 61.28±0.55 | 64.89±0.22 | 67.05±0.26 | 62.39 | 67.10 | 70.27 | 72.57 | 4.5 hr |
| Random | | 57.55±0.52 | 61.82±0.13 | 64.46±0.37 | 66.06±0.27 | 62.47 | 66.10 | 68.93 | 70.55 | 3.7 hr |

MCDO. The reported means and standard deviations were averaged over five trials and one trial for retraining a single model and DE, respectively.

**Effectiveness of fine-tuning.** Tables 1 to 3 compare the AL with SE + FT (Algorithm 2) to baselines. Somewhat surprisingly, SE + FT achieved comparable or even better test accuracies with much shorter runtimes. Fig. 1 shows the progression of test accuracies during the episodes of the SE + FT procedure. The intermediate models, which are only used for acquisitions, exhibit lower accuracies, as expected, since they are fine-tuned and trained with fewer examples for a lower number of epochs. However, at the end of the final episode, where a new classifier is trained from scratch, the test accuracy catches up with the vanilla AL, indicating that even if the accuracies of the intermediate classifiers are inferior, the acquired samples are good enough to obtain a decent classifier at the final episode. We set $S = 5$ for SE + FT, and the reported means and standard deviations were averaged over five trials and one trial for re-training a single model and DE, respectively.

**AL with pretrained models.** As one can see from Table 3, the performance of the methods on Tiny ImageNet is generally poor, presumably due to the complexity of the dataset. To this end, we employed models pretrained on ImageNet (Deng et al., 2009) and examined whether the AL algorithms could fine-tune on pretrained models. We evaluated ResNet-50 and Vision Transformer (Dosovitskiy et al., 2021) backbones and compared SE-based and DE-based AL with VR acquisition. For both backbones, SE outperformed DE significantly, especially with a small number of acquisitions. We

**Table 3:** Test accuracy on Tiny ImageNet according to the ratio of labeled examples

| Acq Fn | Uncertainty | A single model at the final episode | | | | DE at the final episode | | | | |
|---|---|---|---|---|---|---|---|---|---|---|
| ResNet-18 from scratch | | 14% | 15% | 16% | 17% | 14% | 15% | 16% | 17% | Runtime |
| VR | SE | 30.60±0.32 | 31.28±0.16 | 32.38±0.47 | 33.37±0.24 | 36.23 | **37.49** | **38.58** | **39.79** | 14.0 hr |
| | SE + FT | **30.86**±0.41 | **31.77**±0.39 | **32.96**±0.26 | **33.74**±0.31 | **36.32** | 37.13 | 38.19 | 39.01 | 2.3 hr |
| | DE | 30.05±0.21 | 31.20±0.43 | 31.63±0.28 | 32.81±0.65 | 35.83 | 37.17 | 37.97 | 39.34 | 26.1 hr |
| | MCDO | 29.48±0.30 | 31.40±0.89 | 32.35±0.33 | 33.32±0.68 | 35.12 | 37.33 | 38.16 | 38.94 | 14.8 hr |
| Random | | 28.37±0.45 | 28.99±0.44 | 29.04±0.12 | 29.19±0.39 | 32.17 | 33.23 | 32.94 | 32.90 | 8.5 hr |
| ResNet-50 pretrained | | 3% | 5% | 7% | 10% | 3% | 5% | 7% | 10% | Runtime |
| VR | SE | 60.89±0.40 | **66.46**±0.30 | **69.71**±0.42 | **72.31**±0.14 | 62.72 | **68.59** | **71.01** | **73.85** | 3.2 hr |
| | DE | 59.88±0.21 | 61.81±0.43 | 68.21±0.28 | 70.44±0.65 | 62.19 | 67.14 | 70.18 | 73.23 | 5.8 hr |
| Random | | **61.82**±0.34 | 64.11±0.22 | 68.76±0.07 | 70.75±0.16 | **63.40** | 67.59 | 69.64 | 72.09 | 1.8 hr |
| ViT-Base pretrained | | 3% | 5% | 7% | 10% | 3% | 5% | 7% | 10% | Runtime |
| VR | SE | **77.12**±1.10 | **80.68**±2.78 | **85.31**±0.74 | **86.39**±0.14 | **80.51** | **85.81** | **86.95** | **88.73** | 7.1 hr |
| | DE | 69.03±5.47 | 73.20±5.24 | 82.83±1.32 | 86.17±0.26 | 77.01 | 75.92 | 85.86 | 88.06 | 14.9 hr |
| Random | | 71.43±1.03 | 75.98±2.59 | 82.20±0.57 | 84.74±0.92 | 78.73 | 84.50 | 86.83 | 86.87 | 4.9 hr |

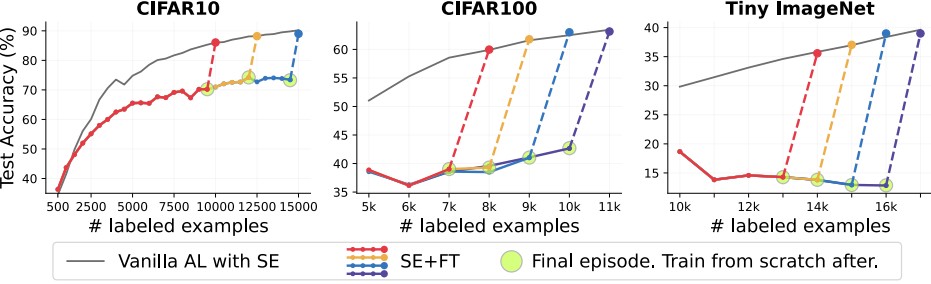

**Figure 1:** Results of vanilla AL (gray) and fine-tuning (FT) at specific episodes (the rest of the colors). CIFAR-10, CIFAR-100 and Tiny ImageNet all used ResNet-18 architecture. Note that before the final episode, the intermediate models show low accuracies.

set $S = 5$ for SE and $S = 3$ for DE, and the reported means and standard deviations were averaged over three trials and one trial for re-training a single model and DE, respectively.

## 5.2 ANALYSIS OF THE UNCERTAINTY ESTIMATION

In this section, we analyze SE in more detail to see why it is more effective than DE for AL. We conjecture that SE builds a committee of models whose predictions are more diverse than the ones from other methods within a single trajectory, and this is a key to its success in AL. To verify this, we measure average KL-divergences and pair-wise disagreement values (Melville and Mooney, 2005) of the predictions computed from SE, DE, and MCDO. The disagreement between two class probabilities $f(\cdot; \boldsymbol{\theta}^{(i)})$ and $f(\cdot; \boldsymbol{\theta}^{(j)})$ on an example $\boldsymbol{x}$ is calculated as

$$d_{i,j}(\boldsymbol{x}) = \mathbb{1}_{\{\arg\max_k f_k(\boldsymbol{x};\boldsymbol{\theta}^{(i)}) \neq \arg\max_k f_k(\boldsymbol{x};\boldsymbol{\theta}^{(j)})\}}. \tag{3}$$

As summarized in Fig. 2 (left), SE generally exhibits much higher KL-divergences and disagreement values among their predictions compared to DE and MCDO. As reported in the literature (Fort et al., 2021), DE shows higher disagreements than MCDO but still much less than SE. We believe that this is mainly due to the nature of AL, where we usually work with a relatively small amount of data and less number of training steps than in typical supervised learning settings. DE parameters are collected at the end of each training run, so the models are likely to reach the local optimum. On the other hand, SE collects the parameter snapshots during a single training run, so some of them may not converge to the local optimum. This degrades the classification accuracy, but as we point out in Section 5.1, for the purpose of acquisition, the diversity in predictions within a single trajectory is more important than the accuracy of the individual models.

Fig. 2 (right) depicts the correlation between the VR values and the predicted probabilities of the ground truth class, along with distributions of VR scores. The test error rate for each bin are shown,

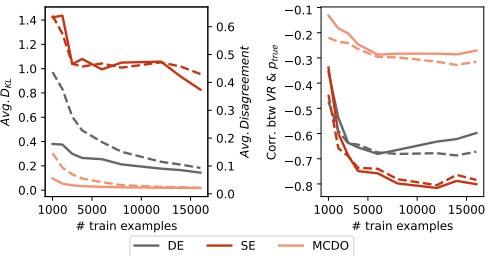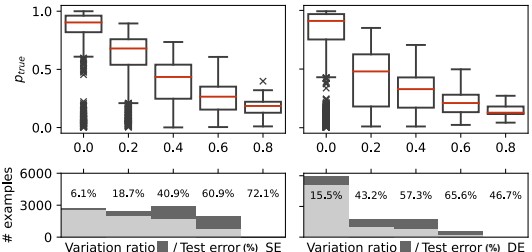

**Figure 2: Left panel**: (Left) average KL-Divergence ▬ and disagreement ▬ between the predictions with different uncertainty estimation methods. (Right) Spearman's rank correlations ▬ and Pearson's correlations ▬ between VR and a predicted probability of the ground truth class $p_{\text{true}}$. **Right panel**: boxplots (up) and histograms (down) of samples binned with the VR values.

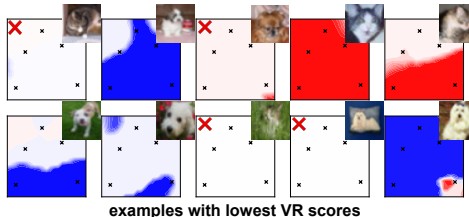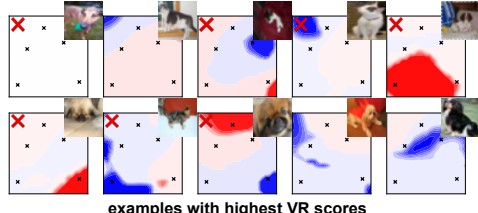

**Figure 3:** Probability maps for the upper right corner images where the blue means dog class and red means cat class. The black small crosses indicate snapshots obtained from SE in the parameter space, while red crosses indicate instances which the model committee predicted incorrectly.

and the number of correctly classified examples is visualized as light gray areas. This can be interpreted as an unnormalized reliability diagram (Murphy and Winkler, 1977). For instance, for the bin with VR = 0.8 (all disagree), SE exhibits a test error of 72.1%, while DE shows only 46.7%. However, for the bin with VR = 0.0 (consensus), SE and DE display error rates of 6.1% and 15.5%, respectively. As examples with high VR scores are initially selected to be labeled, DE is more likely to query examples that the model is already familiar with but has misclassified as confusing.

To provide a qualitative comparison, Fig. 3 shows the map of class prediction probabilities for the committee constructed from SE in the parameter space. The committee members tend to agree on samples with low VR values, whereas they exhibit disagreement on those with high VR values, as expected. This observation underscores the ability of SE to capture parameter snapshots within the same minima while generating diverse predictions on challenging examples with slight changes in the parameter space. Such a phenomenon is closely related to recent advancements in mode connectivity (Garipov et al., 2018) and fast ensembling methods (Huang et al., 2017; Izmailov et al., 2018; Maddox et al., 2019), which reveal that wandering around the wider optima leads to diverse yet more reliable predictions. Based on this empirical evidence, we can conclude that SE is effective at discovering samples with high uncertainties which are predicted to be difficult for the classifier.

## 6 CONCLUSION

In this paper, we demonstrate that estimating uncertainties of the predictions using SE works efficiently for uncertainty-based AL. Through extensive experiments with real-world image classification benchmarks, we empirically confirmed that the AL with SE outperforms AL with DE or MCDO for various choices of acquisition functions. We further presented a novel AL algorithm based on fine-tuning, where we keep a single model and continuously fine-tune it instead of re-initializing the models at the beginning of every episode. The resulting algorithm could achieve comparable classification accuracies given the same number of acquired samples compared to the baselines with much smaller training steps. We provide a detailed analysis of the effectiveness of SE for AL and show that SE builds model committees that yield diverse predictions that are useful for acquiring informative samples.

## REPRODUCIBILITY STATEMENT

We used the Pytorch (Paszke et al., 2019) library in our experiments and algorithms which are described in Algorithm 1 and Algorithm 2. In addition, all experimental details and configurations on hyperparameters are recorded in Appendix B. We will provide an open-source implementation of AL environments and our code of SE and SE + FT algorithms.

## ACKNOWLEDGEMENT

This work was partly supported by Institute of Information & communications Technology Planning & Evaluation (IITP) grant funded by the Korea government (MSIT) (No.2022-0-00184, Development and Study of AI Technologies to Inexpensively Conform to Evolving Policy on Ethics, and No.2019-0-00075, Artificial Intelligence Graduate School Program (KAIST)), and National Research Foundation of Korea (NRF) funded by the Ministry of Education (NRF-2021M3E5D9025030).

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

## A  ADDITIONAL RESULTS AND DISCUSSIONS FOR SE

### A.1  VGG-16

**Table 4:** Comparison between SE and FT with VGG-16 on CIFAR-10

|  | A single model at the final episode | | | | DE at the final episode | | | |
|---|---|---|---|---|---|---|---|---|
|  | 10% | 15% | 20% | 25% | 10% | 15% | 20% | 25% |
| SE | **59.79**$_{\pm 2.36}$ | **68.18**$_{\pm 2.01}$ | **68.88**$_{\pm 1.89}$ | 72.16$_{\pm 1.21}$ | 66.97 | 74.65 | 77.60 | 81.10 |
| SE + FT | 57.78$_{\pm 3.31}$ | 65.22$_{\pm 2.21}$ | 68.58$_{\pm 0.80}$ | **72.50**$_{\pm 1.20}$ | **68.43** | **74.84** | **78.20** | **81.45** |

We also used VGG-16 architecture for fine tuning experiments in Table 4. Somewhat surprisingly, SE + FT has caught up with SE.

### A.2  HYPERPARAMETERS FOR SE

**Table 5:** Test accuracy on different hyperparameters for SE

| # snapshots $S$ | 10% | 15% | 20% | 25% | 30 % |
|---|---|---|---|---|---|
| 5 | **74.28**$_{\pm 0.77}$ | **81.55**$_{\pm 0.38}$ | **86.17**$_{\pm 0.05}$ | 88.34$_{\pm 0.39}$ | 90.09$_{\pm 0.24}$ |
| 10 | 73.71$_{\pm 1.91}$ | 81.05$_{\pm 0.26}$ | 85.65$_{\pm 0.38}$ | 88.60$_{\pm 0.03}$ | 90.06$_{\pm 0.20}$ |
| 20 | 73.49$_{\pm 0.79}$ | 81.27$_{\pm 0.62}$ | 86.04$_{\pm 0.13}$ | **88.68**$_{\pm 0.16}$ | 90.45$_{\pm 0.50}$ |
| 50 | 71.29$_{\pm 0.76}$ | 80.36$_{\pm 0.07}$ | 85.30$_{\pm 0.11}$ | 88.64$_{\pm 0.28}$ | **90.52**$_{\pm 0.11}$ |

| SE learning rate $\eta$ | 10% | 15% | 20% | 25% | 30% |
|---|---|---|---|---|---|
| 0.0001 | 69.66$_{\pm 0.25}$ | 80.25$_{\pm 0.69}$ | 84.95$_{\pm 0.71}$ | 87.42$_{\pm 0.48}$ | 88.85$_{\pm 0.32}$ |
| 0.001 | 71.33$_{\pm 0.87}$ | 79.97$_{\pm 0.19}$ | 85.11$_{\pm 0.02}$ | 88.33$_{\pm 0.21}$ | 89.77$_{\pm 0.04}$ |
| 0.005 | **74.76**$_{\pm 1.15}$ | 81.34$_{\pm 0.07}$ | **85.73**$_{\pm 0.12}$ | **88.87**$_{\pm 0.05}$ | 90.34$_{\pm 0.14}$ |
| 0.01 | 74.49$_{\pm 2.55}$ | **81.95**$_{\pm 0.18}$ | 85.52$_{\pm 0.16}$ | 88.43$_{\pm 0.40}$ | **90.35**$_{\pm 0.20}$ |

| starting point $N_{\text{thres}}$ | 10% | 15% | 20% | 25% | 30% |
|---|---|---|---|---|---|
| 100 | **78.29**$_{\pm 0.05}$ | **82.96**$_{\pm 0.13}$ | 84.89$_{\pm 0.13}$ | 85.66$_{\pm 0.04}$ | 86.08$_{\pm 0.15}$ |
| 125 | 78.02$_{\pm 0.08}$ | 82.74$_{\pm 0.18}$ | 85.33$_{\pm 0.21}$ | 86.65$_{\pm 0.08}$ | 87.43$_{\pm 0.39}$ |
| 150 | 74.49$_{\pm 2.55}$ | 81.95$_{\pm 0.18}$ | 85.52$_{\pm 0.16}$ | **88.43**$_{\pm 0.40}$ | **90.35**$_{\pm 0.20}$ |
| 175 | 74.65$_{\pm 1.43}$ | 82.09$_{\pm 0.44}$ | **85.66**$_{\pm 0.16}$ | 88.14$_{\pm 0.37}$ | 89.75$_{\pm 0.61}$ |
| Random | 74.74$_{\pm 0.05}$ | 80.64$_{\pm 0.71}$ | 83.91$_{\pm 0.55}$ | 86.00$_{\pm 0.07}$ | 87.53$_{\pm 0.24}$ |

Table 5 summarizes various settings of SE hyperparameters according to the proportion of labeled examples. Here, we only compare test accuracies of a single model trained with vanilla SGD on CIFAR-10 dataset with VR acquisition function. We additionally applied Stochastic Weight Averaging (SWA) (Izmailov et al., 2018) when training from scratch at the end to be less sensitive to hyperparameter settings and effectively compare the quality of queried examples.

**Number of snapshots $S$.**  One may increase the number of snapshots collected to better acquire examples to be labeled since SE does not incur additional training costs. However, collecting more snapshots linearly increases the inference time for AL due to multiple forward passes. Overall, there is no significant difference in performance gains from the increase in the number of snapshots. Increasing the number of snapshots results in lower performance than random at the beginning, but performance tends to increase as episodes continue. Based on these findings, one might come up with a strategy that uses a small number of snapshots at first and then increases the number of snapshots in latter episodes. All results used SE with VR acquisition function and were averaged over two trials including the ones with random acquisition. We used as the same hyperparameter settings described in Appendix B. Here, the total epochs $N$ and SE staring epoch $N_{thres}$ are fixed to 200 and 150, respectively. The jump between snapshots differs accordingly.

**SE learning rate.** In Algorithm 1 and Algorithm 2, a learning rate is adjusted by the learning rate scheduler $\eta(.)$, and we used a high constant learning rate while collecting snapshots except for CIFAR-100 due to the instability of training. We maintain the same learning rate for CIFAR-100. The high constant learning rate used when collecting snapshots contributes to diverse predictions and, consequently, the success of SE in the AL context. The learning rate during training was all fixed to 0.001. When the SE learning rate value was too small ($\eta = 0.0001$), the performance was inferior. However, too high learning rate may cause a model to diverge and move to different mode values or meaningless areas in the weight space, which surely degrades the acquisition quality. We also include the previous result with CIFAR-100 dataset in Table 9, where we used $\eta = 0.005$ during SE. In our preliminary experiments, the VR acquisition function showed robustness despite the decrease in the accuracy of the model's predictions because it uses the count of members who agree rather than predicted probabilities, whereas other acquisition functions did not perform well. Here, the number of snapshots $S$ and SE starting epoch $N_{thres}$ are fixed to 5 and 150, respectively. The total number of epochs $N$ was also fixed as 200.

**The starting point of SE $N_{thres}$.** We tried four different starting points of SE, or burn-in time, $N_{thres}$. This experiment shows how important it is to collect snapshots after the model sufficiently converges and why previous methods have failed. For example, Beluch et al. (2018) collected snapshots from the beginning of the training (like at 40, 80, 120, 160, and 200 epoch). We showed that when snapshots were obtained before the model sufficiently converged ($N_{thres} = 100, 125$), it performed even worse than the random acquisition. Similarly, when the jump between snapshots was too small (in case of $N_{thres}$=175, and therefore jump=5), the performance dropped in later episodes. Here, the number of snapshots $S$ and SE learning rate were fixed to 5 and 0.01, respectively. The total number of epochs $N$ was also fixed as 200.

A.3  SE WITH FINE-TUNING HYPERPARAMETERS

SE + FT algorithms are governed mainly by the two hyperparameters: regularization hyperparameter and replay buffer size.

**Table 6:** Test accuracy on different regularization hyperparameter $\lambda$ for FT

| lambda $\lambda$ | 10% | 15% | 20% | 25% | 30% |
|---|---|---|---|---|---|
| 0.0 | **77.66**±0.10 | 82.45±0.22 | 86.01±0.43 | 87.70±0.36 | 89.05±0.75 |
| 0.001 | 76.86±0.40 | 82.83±0.18 | 86.27±1.76 | **88.12**±0.19 | 88.45±0.14 |
| 0.005 | 76.99±0.10 | 82.70±0.05 | 86.24±0.22 | 87.79±0.06 | 89.12±0.22 |
| 0.01 | 77.09±0.02 | **82.86**±0.13 | 85.78±0.59 | 87.79±0.16 | **89.56**±0.22 |
| 0.02 | 73.91±2.02 | 82.74±0.72 | **86.30**±0.25 | 87.60±0.74 | 82.14±8.42 |

**Regularization hyperparamter $\lambda$.** The regularization hyperparamter $\lambda$ has a role of controlling the balance between maintaining a single trajectory from the previous episode and adapting the model to newly acquired samples. It is crucial to find appropriate $\lambda$ values for $\ell_2$-regularizer $\|\boldsymbol{\theta} - \boldsymbol{\theta}_0\|^2$. Table 6 shows test accuracies at 10, 15, 20, 25, and 30 episode (with 5K, 7.5K, 10K, 12.5K, and 15K labeled examples, respectively) depending on different $\lambda$ values on CIFAR-10 dataset. Here, the replay buffer size was fixed as 2,500. Although the acquisition quality is improved with regularization, there was no substantially outstanding $\lambda$ value throughout the entire episodes. Therefore, in the experiments in Section 5, the $\lambda$ value was fixed as 0.01.

**Table 7:** Test accuracy at episode 30 according to the replay buffer size on CIFAR-10

| Replay buffer size | 500 | 1,000 | 1,500 | 2,000 | 2,500 | 3,000 |
|---|---|---|---|---|---|---|
| Acc (%) | 87.79 | 88.26 | 88.57 | 89.30 | **89.39** | 88.91 |

**Replay buffer size.** In the case of the number of data used for fine tuning process, some of the data labeled in the previous episodes were added to the newly acquired data. For CIFAR-10, we

used the budget size for 500. Moreover, we add additional 2,000 randomly sampled examples from the previous labeled data. For both CIFAR-100 and Tiny ImageNet, we add 1,000 random sampled from labeled data to 1,000 newly acquired data. This has significantly reduced the training cost. Table 7 shows test accuracies of the final model (trained from scratch) with 15,000 labeled examples according to the size of replay buffer in each episode.

## A.4 Visualizations

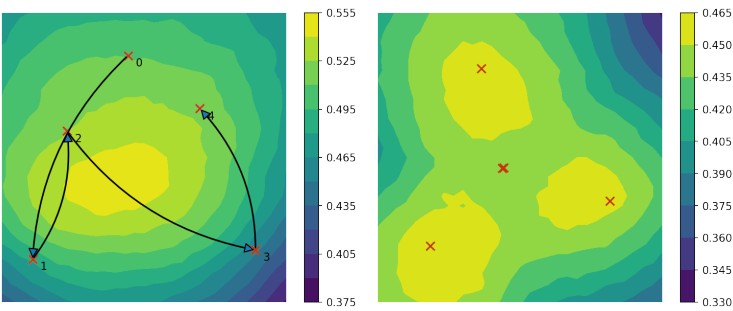

**Figure 4:** Loss surface on test set of SE (left) and DE with same initialization (right) for the visualization purpose in the parameter space when trained with the first 2,000 examples of CIFAR-10 dataset. Contours represent test accuracy, and red points denote weights gathered for AL.

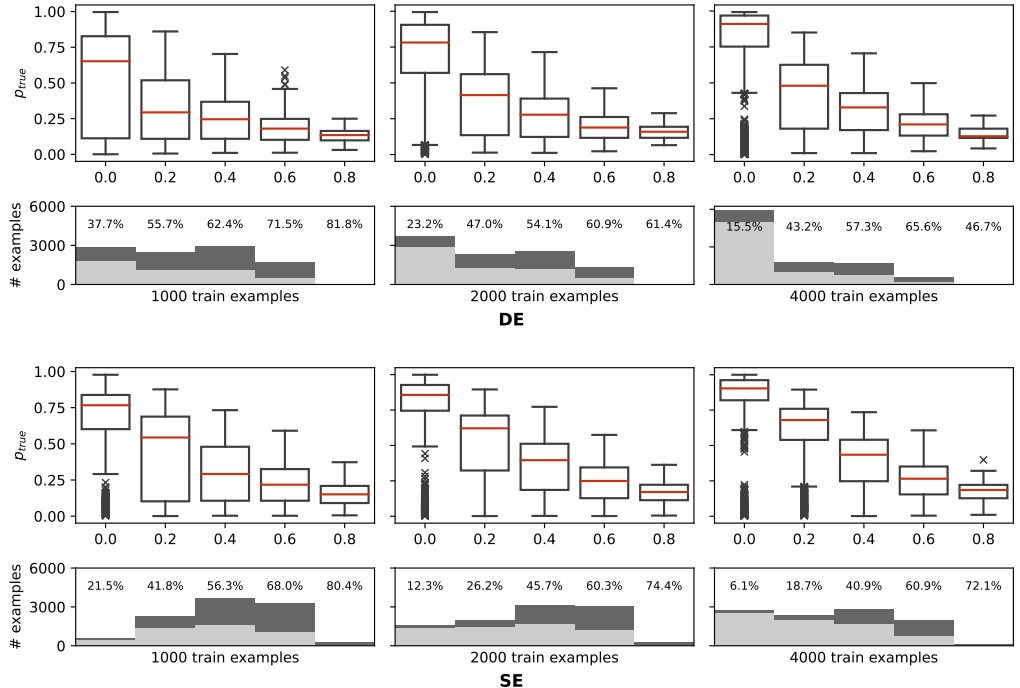

**Figure 5:** More boxplots and histograms for DE (up) and SE (down) plotted in the same manner as Fig. 2 (right), showing the tendency as the number of labeled examples increases.

Fig. 4 shows test accuracy map in the parameter space, and red crosses indicate parameters collected for ensemble. Note that two contours have different color scales. With a high learning rate, SE picks snapshots that are weak individually but strong together around the wider optimum. Each member of DE falls into their own narrow optima.

Similar to Fig. 2 (right), Fig. 5 shows the correlation between the VR values and the predicted probabilities of the ground truth class, along with a histogram of VR values. In the histogram, the percentages denote the test error for each bin, and light gray areas depict the number of examples which the committee predicted incorrectly. Fig. 5 clearly illustrates that DE tends to exhibit overconfident results. Even when the number of acquired data increases, the degree of overconfidence by DE remains severe. However, VR scores calculated with SE showed much better correspondence with the actual error rate. Here, we trained the model the first {1000, 2000, 4000} CIFAR-10 examples in the train set.

## B  EXPERIMENTAL DETAILS

### B.1  BASELINE DESIGN

When conducting our experiments, we placed great importance on achieving robustness, reproducibility, and generalizability. Our experiments on the CIFAR-10 and CIFAR-100 datasets yielded average test accuracy scores of 90.15% and 68.02%, respectively, over five trials. We achieved these results using the VR acquisition function and acquiring 30% of the labels (equivalent to 15,000 examples) under the settings outlined in Appendix B.3. In contrast, a recent survey by Munjal et al. (2022) reported baseline results of 90.87% and 59.36% test accuracy with 40% random samples.

In order to facilitate reproducibility of our results, we provide not only the code and configurations, but also the indices of the queried examples. Reported runtimes are based on an Ubuntu 20.04 server with an AMD Ryzen-4 5900X CPU and 64GB RAM, as well as an NVIDIA RTX-3090 GPU with 24GB VRAM. For faster training on the Tiny-ImageNet dataset, we additionally employed FP16 (Automatic Mixed Precision).

### B.2  ACQUISITION QUALITY

To ensure a fair and objective comparison of acquisition quality, we devised an experiment in which the data selected by three different methods, namely SE, DE, and MCDO, were used to re-train both a single model and an ensembled model. By comparing the effectiveness of the selected data in learning, we aimed to provide a comprehensive evaluation of the performance of each method. In cases where multiple experiments were conducted, we carefully examined the results to ensure that they exhibited a similar trend across experiments. We randomly selected one experiment and reported the re-trained results obtained from that experiment due to the limited resources.

### B.3  HYPERPARAMETERS

**Table 8:** Summary of hyperparameters

| Dataset | Architecture | Optimizer | Base lr | Momentum | Weight decay | Scheduler | SE lr | Max epoch | SE epochs | # snapshots |
|---|---|---|---|---|---|---|---|---|---|---|
| CIFAR-10 | ResNet-18 | SGD | 0.001 | 0.9 | 0.01 | ONECYCLE | 0.01 | 200 | 50 | 5 |
| | VGG-16 | SGD | 0.001 | 0.9 | 0.01 | ONECYCLE | 0.01 | 100 | 100 | 5 |
| CIFAR-100 | ResNet-18 | SGD | 0.001 | 0.9 | 0.005 | ONECYCLE | 0.01 | 200 | 50 | 5 |
| Tiny ImageNet | ResNet-18 | SGD | 0.001 | 0.9 | 0.0001 | Steps | 0.01 | 100 | 50 | 5 |
| | ResNet-50 | SGD | 0.001 | 0.9 | 0.0001 | ONECYCLE | 0.05 | 50 | 25 | 3 |
| | ViT-base-224 | SGD | 0.001 | 0.9 | 0.0001 | Constant | 0.05 | 50 | 25 | 3 |

We used a standard SGD optimizer with the following hyperparameters for both CIFAR-10 and CIFAR-100 datasets: a base learning rate of 0.001, momentum of 0.9, and weight decay of 0.01. The mini-batch size was set to 64 for CIFAR-10 and 128 for CIFAR-100. During SE, we raised the learning rate to 0.01 for CIFAR-10 or dropped it to 0.0001 for CIFAR-100. In our preliminary experiments, we also tried increasing the learning rate to 0.005 during SE for CIFAR-100, but the results (shown in Table 9) were not as good as those obtained with the final settings. Our results showed that SE with VR outperformed DE and MCDO, but other acquisition functions did not perform well with SE. Using a learning rate lower than the base learning rate can help collect snapshots yielding decent predictions if the training itself is unstable. In contrast, a learning rate higher than the base learning rate worked well in other hyperparameter settings and datasets.

**Table 9:** Previous results on CIFAR-100

| Acq Fn | Uncertainty | A single model at the final episode | | | | DE at the final episode | | | | Runtime |
|--------|-------------|------|------|------|------|------|------|------|------|---------|
| | | 16% | 18% | 20% | 22% | 16% | 18% | 20% | 22% | |
| VR | SE | $59.34_{\pm 0.36}$ | $61.02_{\pm 0.11}$ | $\mathbf{63.02}_{\pm 0.23}$ | $64.14_{\pm 0.27}$ | 64.26 | 66.21 | **68.27** | 69.24 | 7.2 hr |
| | SE + FT | $\mathbf{59.56}_{\pm 0.58}$ | $\mathbf{61.17}_{\pm 0.37}$ | $62.88_{\pm 0.26}$ | $\mathbf{64.81}_{\pm 0.27}$ | **64.35** | **66.23** | 67.68 | **69.72** | 0.8 hr |
| | DE | $58.17_{\pm 0.20}$ | $60.44_{\pm 0.21}$ | $62.08_{\pm 0.29}$ | $63.46_{\pm 0.35}$ | 63.78 | 65.99 | 67.99 | 69.22 | 28.7 hr |
| | MCDO | $58.81_{\pm 0.34}$ | $60.62_{\pm 0.26}$ | $62.38_{\pm 0.26}$ | $63.50_{\pm 0.17}$ | 63.23 | 65.09 | 66.80 | 67.69 | 6.2 hr |
| Random | | $58.93_{\pm 0.20}$ | $60.58_{\pm 0.29}$ | $61.95_{\pm 0.24}$ | $62.80_{\pm 0.44}$ | 63.63 | 65.09 | 66.68 | 67.82 | 5.7 hr |

To speed up convergence and reduce the effort of finding optimal hyperparameters, we used One Cycle learning rate scheduler (ONECYCLE) proposed by Smith and Topin (2019), setting max_lr to 0.01, for both datasets. For augmentations, we normalized images with the mean and variance of all images in the train set and applied random horizontal flip to both datasets. Additionally, random cropping was applied to CIFAR-100. We trained the models for a total of 200 epochs, and for SE, we collected five snapshots with an additional 50 epochs (10-epoch interval).

**Table 10:** Performance gain with ONECYCLE scheduler with 10,000 labeled examples on CIFAR-10. The differences from random acquisition are in parentheses.

| Acq Fn | w/ ONECYCLE | w/o ONECYCLE | $\Delta$ |
|--------|-------------|--------------|----------|
| VR | 86.0% ($\Delta$ 1.9%p) | 84.1% ($\Delta$ 2.9%p) | 1.9%p |
| BALD | 85.4% ($\Delta$ 1.3%p) | 83.4% ($\Delta$ 2.2%p) | 2.0%p |
| ME | 85.2% ($\Delta$ 1.1%p) | 84.1% ($\Delta$ 2.9%p) | 1.1%p |
| Random | 84.1% | 81.2% | 1.9%p |

We also evaluated the effect of using ONECYCLE on the performance of various acquisition functions for CIFAR-10 with ResNet-18 in Table 10. Without ONECYCLE, the performance gain compared to random acquisition of all acquisition functions decreased, with the differences ranging from 2.2%p to 2.9%p. However, when using ONECYCLE, the performance of all acquisition functions improved, but the differences from random acquisition decreased to a range of 1.1%p to 1.9%p. Despite the fact that the differences from random sampling were lower with ONECYCLE than without ONECYCLE, we chose to report the results with ONECYCLE as they are more relevant for practical applications with limited labeled examples. The reported accuracies are averaged over five trials.

For Tiny-ImageNet dataset with ResNet-18, we also used a SGD optimizer with momentum of 0.9 and weight decay of 0.0001. We adjusted the learning rate to 0.1 for the first half of the training epochs, 0.01 until the 75% of the training epochs, and 0.001 for the rest. During SE, the learning rate is increased to 0.05. For augmentations, we used random crops and random horizontal flips, which are the *de-facto* standard random augmentation strategies when reporting in academia. To reduce the computational overhead in ensemble-based acquisitions due to multiple forward passes of a large unlabeled pool $\mathcal{U}$, we first randomly draw $Q$ unlabeled examples from the pool and measured the scores, following Beluch et al. (2018) and Yoo and Kweon (2019). We set $Q$ to 10,000. The total number of training epochs is 100, and for SE we collected five snapshots with additional 50 epochs (10 epoch interval).

For transfer learning experiments on Tiny ImageNet, we used pretrained weights for ResNet-50 from Torchvision (Paszke et al., 2019) and those for ViT-base-224 from PyTorch Image Models (Wightman, 2019) and replaced the final linear classification head. Instead of using the original image size, an image was scaled up to $224 \times 224$ resolution to match the model trained with ImageNet dataset. The total number of training epochs is 50, and for SE we collected 5 snapshots with additional 25 epochs (5 epoch interval). We here also used a SGD optimizer with momentum of 0.9 and weight decay of 0.0001. For ResNet-50, we used ONECYCLE (the same as above) and for ViT-base, we did not use a learning rate scheduler (constant learning rate). We also set $Q$ to 10,000. Please see Table 8 for a summary of hyperparameters.

### B.4 MODEL STRUCTURES

For all experiments above, the structure of ResNet-18 model is slightly modified in order to fit $32 \times 32$ and $64 \times 64$ images following the standard protocol as follows:

- The kernel size of the first convolution layer (`conv1`) has changed to 3, and the stride is changed to 1.
- The max pooling layer is disabled.
- For MCDO, a dropout layer with dropout rate $p = 0.5$ is added in front of the final linear classifier layer, since the original implementation has no dropout layers.

Similarly, the structure of VGG-16 model is slightly modified as follows:

- The batch normalization layer is added next to every convolution layer.
- The average pooling layer is disabled.
- No additional dropout layer is attached to the model since its classifier already has two dropout layers with dropout rate $p = 0.5$. We turned on the dropout layer when querying with MCDO.

These additional changes including MCDO layers had no effects on all reported figures in Section 5 since we reported acquisition qualities by re-training models with the same structure with the examples queried by each method.

