# OpenReview forum: "A Simple Yet Powerful Deep Active Learning With Snapshots Ensembles"
_ICLR.cc/2023/Conference — ICLR 2023 poster_

### Official Review · Reviewer_qYVA · 2022-10-18

**Confidence:** 4
**Correctness:** 3
**Technical Novelty And Significance:** 3
**Empirical Novelty And Significance:** 4
**Recommendation:** 8

**Clarity, Quality, Novelty And Reproducibility:**

- This paper is quite clear.
- Figure 3 is a bit hard for me to understand. Perhaps cut it, or add more explanation.
- The right side of Figure 2 is a bit mysterious. What are the percentage values in the bottom?
- The experimental work is original, though all techniques appear in previous work (not necessarily in active learning).

**Strength And Weaknesses:**

Strengths:
 - The combination of techniques used in the experiments dramatically improves runtime with equal or better runtime.
 - The observation that warm-starting works for data selection (though not for the final model) is very interesting and important.
 - This paper's combination of recent ideas in parameter sampling with old ideas in active learning is very timely.


Weaknesses:
 - It seems that some rows are missing for the experiments. For example, for CIFAR10, BALD with SE+FT, ME with MCDO, and ME with SE+FT are missing. Furthermore, it appears that MCDO and SE+FT are missing for the pre-trained models.
 - I'm curious why margin uncertainty sampling (best vs second best or BvSB) is not used. In the literature, this method usually outperforms entropy uncertrainty sampling.


**Summary Of The Paper:**

This paper combines two ideas: snapshot ensembles and sampling-based (uncertainty and mutual information) active learning. Snapshot ensembles are shown to empirically perform better than standard (deep) ensembles and Monte Carlo dropout, when used in conjunction with uncertainty sampling (and BALD). Furthermore, warm-starting (instead of training from scratch) is shown to work well empirically for data collection with active learning as long as the model is evaluated with a model trained from scratch.

**Summary Of The Review:**

While "only" combining existing ideas, this paper experimentally finds two very practically useful observations that I think the research community will appreciate.

---

> ### Author Response · Authors · 2022-11-14
> **Thank you for your review**
>
> We appreciate your time in reviewing our paper and your constructive comments.
>
> ### Figures
>
> Thank you for your suggestion about the Figures. Please refer to the overall comment for the errata. Your suggestions make the paper look much more readable and easy to follow.
>
> ### Missing experiments
>
> We did additional experiments on the rows that were missing on CIFAR10. In addition, we provided BALD and Max Entropy results on CIFAR100. In the case of experiments about pretrained models on Tiny ImageNet, we used only 50 epochs for training for each episode. Since this is similar to FT in terms of computational cost, no additional experiments were conducted. Also, MCDO experiments require modification for the model, but it is not straightforward and may result in a bias toward our results.
>
> ### Margin uncertainty sampling
>
> Our contribution is to show the performance of SE compared to DE in AL, and we chose three acquisition functions based on epistemic uncertainty. As far as we understand, the margin-based sampling method uses the mean probability of ensemble members to compare the best and the second-best predicted probabilities, and multiple models are not necessary to calculate the scores. Although we are aware that margin-based sampling works well in the AL literature, it does not directly take epistemic uncertainty into account. Therefore, we did not consider margin-based sampling as our priority. Nonetheless, our algorithm is broadly applicable to any uncertainty-based acquisition functions.

---

> > ### Comment · Reviewer_qYVA · 2022-11-14
> > **Reviewer response**
> >
> > Thank you for your response. I just have a couple comments about the use of the term "epistemic uncertainty".
> >
> > I agree with Reviewer s1VV that maximum entropy and variation ratio are not actually epistemic uncertainty. Instead, only BALD is epistemic uncertainty. Furthermore, the paper's definition of maximum entropy is simply entropy-based uncertainty sampling using averaged ensemble probabilities. The same could be done for margin-based uncertainty sampling. These aren't major points, but I think the paper should use standard terminology to avoid confusion. A quick fix would be replacing "epistemic uncertainty" with "uncertainty" except possibly in the case of BALD.

---

> > > ### Author Response · Authors · 2022-11-15
> > > **Thank you for your review**
> > >
> > > Thanks again for your feedback on the terminology to avoid confusion. We uploaded a revision by replacing “epistemic uncertainty” with “uncertainty” except for BALD.
> > >
> > > * Section 2.2: Replace “epistemic uncertainty” with “uncertainty”
> > > * Section 2.2: Rewrite a sentence stating that BALD measures epistemic uncertainty.

---

> > > > ### Comment · Reviewer_qYVA · 2022-11-15
> > > > **Reviewer response**
> > > >
> > > > Thank you for the changes. However, I think the issue goes beyond Section 2.2. I found the word "epistemic" 9 places in revision where most don't make sense in my opinion.
> > > >
> > > > To clarify, my understanding is that the usual terminology is that uncertainty without any modifiers includes both epistemic and aleatoric uncertainty. This "total" uncertainty is a function of a discrete probability distribution (like a conditional label distribution) and can be made concrete by measures such as entropy, probability of non-mode label (least confident), or margin. Aleatoric uncertainty is usually the uncertainty of the best-in-class model or for the true label distribution (i.e., even if you knew the true model of the world, what uncertainty is still present). Epistemic is typically defined as the total uncertainty minus the aleatoric uncertainty, highlighted in the BALD equation: "mutual information" = "entropy" - "conditional entropy" or equivalently "epistemic uncertainty" = "total uncertainty" - "aleatoric uncertainty".
> > > >
> > > > Please let me know if any part doesn't make sense. If desired, I can hunt up references to back up this terminology. Also, for the other reviewers, please let me know if this is not your understanding.

---

> > > > > ### Author Response · Authors · 2022-11-16
> > > > > **Thank you for your review**
> > > > >
> > > > > We sincerely thank you for your concerns and efforts to further enhance the quality of our paper.
> > > > >
> > > > > The “epistemic” uncertainty we are referring to is the uncertainty in the model (or its parameter $\theta$). We are borrowing the terms epistemic and aleatoric uncertainties from the seminal paper [Kendall and Gal, 2017], where they describe
> > > > >
> > > > > * Aleatoric uncertainty captures noise inherent in the observations (~ data uncertainty).
> > > > > * Epistemic uncertainty accounts for uncertainty in the model parameters (~ model uncertainty).
> > > > >
> > > > > So if we assume the decomposition of the total uncertainty = aleatoric + epistemic, then as you said the epistemic uncertainty would be uncertainty except for aleatoric.
> > > > >
> > > > > In an active learning scenario, we have a set of class probability vectors computed from multiple $\theta$s. The aleatoric uncertainty would be the uncertainty in the class probability vector itself; the uncertainty of the categorical distribution defined with each class probability vector. The epistemic uncertainty, on the other hand, would be the uncertainty due to the presence of multiple class probability vectors computed from multiple parameters; we can either collect multiple parameters via snapshot ensembles, deep ensembles, or even MCMC. The term epistemic uncertainty being mentioned in our paper refers to this concept - the uncertainty of class probability vectors (from a committee of models) arising from the uncertainty in the model parameter $\theta$.
> > > > >
> > > > > If you still are concerned with the term epistemic due to the confusion that may arise from it, how about changing the term to “model uncertainty”, which would in our opinion be more straightforward to interpret?
> > > > >
> > > > > [Kendall and Gal, 2017] What Uncertainties Do We Need in Bayesian Deep Learning for Computer Vision? NIPS, 2017.

---

> > > > > > ### Comment · Reviewer_qYVA · 2022-11-19
> > > > > > **Epistemic Uncertainty Discussion**
> > > > > >
> > > > > > I read the intro of Kendall & Gal (2017). To quote: "epistemic uncertainty accounts for uncertainty in the model parameters – uncertainty which captures our ignorance about which model generated our collected data." I think my definition of epistemic uncertainty is in line with that paper.
> > > > > >
> > > > > > I agree with your statement: "The aleatoric uncertainty would be the uncertainty in the class probability vector itself; the uncertainty of the categorical distribution defined with each class probability vector." But I think what you are calling the epistemic uncertainty is actually the total uncertainty. To describe this, consider the following scenario:
> > > > > >
> > > > > > Suppose we have a binary classification task. We have collected a lot of data, so when we sample several $\theta_i$, they are similar. In particular, for a fixed input $x$, the class probability vectors induced on $x$ by the parameters are:
> > > > > >
> > > > > > $\theta_1: (0.49,0.51)$
> > > > > >
> > > > > > $\theta_2: (0.52,0.48)$
> > > > > >
> > > > > > $\theta_3: (0.5, 0.5)$
> > > > > >
> > > > > > $\theta_4: (0.48, 0.52)$
> > > > > >
> > > > > > $\theta_5: (0.49, 0.51)$
> > > > > >
> > > > > > I think you would agree that there is little epistemic uncertainty here, we have a lot of data and have a very good idea of what the true $\theta$ is. There is uncertainty, but it is almost all aleatoric. If we knew $\theta$ exactly, the probability class vector would be something like $(0.5,0.5)$.
> > > > > >
> > > > > > However, the maximum entropy criteria in the paper (and variation ratio) would be high for this example. Thus, although the maximum entropy is computed from samples, it is not the epistemic uncertainty (recall this is very small for this example), but rather the total uncertainty. In a sense, by averaging multiple $\theta$'s, you are getting better uncertainty estimates via ensembling, but it is still total uncertainty.
> > > > > >
> > > > > > I might consider writing the paper around "uncertainty-based acquisition functions", "uncertainty", or something like that, rather than "epistemic uncertainty ". Please let me know your thoughts.

---

> > > > > > > ### Author Response · Authors · 2022-11-19
> > > > > > > **Thank you for your review**
> > > > > > >
> > > > > > > Thanks for your valuable feedback and patients.
> > > > > > >
> > > > > > > Of course, as you said, the uncertainty modeled with the acquisition functions (ME or VR) are total uncertainties; what we intended by the keyword "epistemic uncertainty" is the uncertainty of the parameters $\theta$, not the acquisition functions themselves. Given a fixed likelihood $p(y|x,\theta)$, we have several options to choose $\theta_1, \theta_2, ... \sim p(\theta | \text{data})$. So that is the reason why we mention that we estimate "epistemic uncertainty" either via snapshot ensemble, deep ensemble, or MCMC. However, as you also pointed out, once we mention the committees of predictions or their disagreements, we are already mentioning the mix of both aleatoric (uncertainty from $p(y|x,\theta)$) and epistemic (uncertainty in $\theta$).
> > > > > > >
> > > > > > > Hence we agree with your point and changed the keywords epistemic uncertainty to uncertainty throughout the article. Many thanks for your comments clarifying the confusion around those concepts.

---

### Official Review · Reviewer_s1VV · 2022-10-19

**Confidence:** 4
**Correctness:** 4
**Technical Novelty And Significance:** 3
**Empirical Novelty And Significance:** 3
**Recommendation:** 6

**Clarity, Quality, Novelty And Reproducibility:**

The paper is well written and the approach has novelty and the significant details of the approach are presented so that the paper appears reproducible.

I may quibble that some of the maximum entropy and variation ratio are not selecting the samples with highest epistemic uncertainty. In these cases, this reviewer believes these are estimates of aleatoric uncertainty while BALD is indeed a measure of epistemic uncertainty.  Nevertheless, these issues are not central to the main result of the paper which is the use of SE rather than DE.

This reviewer would like to see more assessment of the reduction in run-time via SE+FT over SE. This reviewer would like to see some discussion of selection of the SE specific hyperparameters.

It also seems that what really matters is the runtime to select the next m samples to label rather than the total runtime?  What really matter is that this runtime is comparable (less than) the time for the annotators to actually label the batch of m samples.

In the bottom paragraph of page 6, SE and DE are discussed in relation to baselines.  What are these baselines?

The table captions should explain what is being shown.  Are the results the probability of correct classification?  Are the percentages in the column header the percentage of unlabeled data that are eventually labeled?

I could not see where Figure 1 is referenced in the narrative.

I did not follow Figure 3.  It seems that the upper left in lowest VR and highest VR scores is equally confused between cat and dog for all 5 snapshots. In that case, it would seem that max p would be comparable.  If so, why the discrepancy between VR scores. I am also not clear what the probability map actually represents.

Nevertheless, I do like how the paper tries to show how the SE method provides ensembles with more diverse opinions.

It would seem that MCMC methods such as Langevin dynamics would provide more diverse ensembles in a single pass than SE methods.  Perhaps some discussion of why this might not be the case or if this should be explored as future work would seem appropriate.




**Details Of Ethics Concerns:**

No ethical concerns.

**Strength And Weaknesses:**

The strength of the paper is that is well written, presents a reasonable somewhat novel approach for active learning, and the experimental results do show the potential of the approach.  The weakness of the paper is that both the deep ensemble and snapshot ensemble approaches are controlled by many various hyperparameters (e.g. number of samples, burn in time N_thresh, jumps between snapshots J, etc.).  These particular hyperparameters do not seemed to be discussed in the context of any hyperparameter search method in the paper or the appendices.  Choosing these hyperparameters judiciously seems important to appreciate the reduction in runtime. Also, the SE with fine tuning seems to be a significant advance over SE, but it is not fully evaluated in the experimental results.  It seems that the authors ran out of time to run the experiments, and the results are incomplete.

**Summary Of The Paper:**

This paper proposes a more time efficient active learning method based upon epistemic uncertainty by incorporating snapshot ensembles rather than using the traditional method of deep ensembles. The experimental results demonstrate comparable recognition performance with a significantly lower runtime.

**Summary Of The Review:**

The work appears novel and the experiments demonstrate the effectiveness of the method. The results seem somewhat incomplete at the SE+FT seems like the significant results that could be explored more.

---

> ### Author Response · Authors · 2022-11-14
> **Thank you for your review**
>
> Thank you for reviewing our paper and your constructive comments. We appreciate your suggestions about figures, tables and where the specific statement about baseline is missing. Please refer to the overall comment for the errata. Your suggestions make the paper look much more readable and easy to follow.
>
> ### Additional experiments on hyperparameters
>
> The results in the initial submission is based on the experimental settings reported in previous works. We additionally provided results under various hyperparameter settings in appendix A (different learning rates, number of acquisitions of each episode, and number of ensemble members). Below, we present some analysis on our findings for each hyperparameter. Overall, we argue that our algorithm is not extremely sensitive to the choice of hyperparameters, and using decent values is sufficient to get reasonable performances.
>
> **Number of snapshots** –
>
> Table 5 presents the performance with respect to the different numbers of snapshots being used for acquisition. We find that increasing the number of snapshots did not necessarily result in better performance, especially in the early stages. In practice, we suggest using a moderate number of snapshots, considering the computational budget.
>
> **Different learning rates** –
>
> Table 5 also presents results showing the effect of learning rates for snapshot ensembling. There is a tradeoff between diversity and accuracy due to the choice of learning rates; too large learning rates would encourage diversities in predictions, but it may cause models to diverge, but too small learning rates would produce predictions that are not diverse enough.
>
> **SE Starting point** –
>
> In SE, it is important to obtain snapshots after the model reaches the basins of attractions of local minima. If the snapshots are obtained before the model sufficiently converges, the performance may be good in the early stages of AL, but the performance will eventually be degraded in later stages.
>
> ### Runtime
> Runtimes shown in the Table 1 to 3 are mainly to explain relative computational costs for both training and calculating acquisition functions. In CIFAR-10, for example, the number of training epochs required for each uncertainty method is 200, 200+50, 200*5=1000 for MCDO, SE, and DE, respectively. The additional inference steps needed are 25, 5, and 5 for MCDO, SE, and DE, respectively. Random acquisition does not require inference time. Here, we ignored CPU runtime (mainly used for calculating acquisition functions) since it is negligible compared to GPU time. The specification of the machine we used is recorded in Appendix B.
>
> As you clearly mentioned, it would be ideal that the training time matches the time needed for annotators to label m batch samples. However, the runtime of each episode even varies depending on the number of labeled examples. Also, the hyperparameters such as the budge size, the number of episodes, and etc. should be determined by practitioners considering their own situations. Nevertheless, SE and SE+FT reduce overall runtime while outperforming DE, thus giving them more freedom in AL environments.
>
>
> ### Discussion on SE and FT
>
> To further analyze FT, we did additional experiments about average KL-Divergence and disagreement between the predictions with DE, SE, and FT with varying training set sizes (Figure 2, (Left)). Remarkably, FT initially had a very low KL divergence value (about 0.078 at train set size 6000) and grew significantly from the time that the train set reached 8000. (about 0.33 at training set size 8000) And this value is higher than DE and lower than SE. That means, FT exhibits a diversity higher than DE but lower than SE. Considering that FT even outperforms SE, this may imply that FT finds achieving a better trade-off between accuracy and diversity compared to SE or DE.
>
> ### MCMC as an uncertainty estimate
> Thanks for your suggestion, indeed, an active learning with stochastic gradient MCMC (SGMCMC) method based on cyclical learning rate [Zhang et al., 2019, Cyclical Stochastic Gradient MCMC for Bayesian Deep Learning] would be similar to a Bayesian version of our algorithm. We leave it as future work.

---

### Official Review · Reviewer_qtq1 · 2022-10-21

**Confidence:** 4
**Correctness:** 4
**Technical Novelty And Significance:** 4
**Empirical Novelty And Significance:** 3
**Recommendation:** 8

**Clarity, Quality, Novelty And Reproducibility:**

The paper is clear, and novel in the sense that it points out SE should be carried out with the final few episodes retrained from scratch. Reproducibility is good with enough details disclosed.

**Strength And Weaknesses:**

Strength:

The idea is simple. Combining SE to AL is not a new idea as I believe a lot of AL practitioners (like myself) training multiple models (DE) would think of possibly training less, like using FT. The problem is majority of us stop short when we see the fine-tuned accuracy is much lower than retraining from scratch, not realizing that the final accuracy is what we want....

Just like the title, this simple idea is powerful as demonstrated in the experiments. The paper is well-written and easy to understand. The details are sufficient for the readers to follow and reproduce the results.

Also, just want to point out that Figure 1 is crucial for helping the reader as it conveys an important point that before the final episode, the model is at low accuracy. I couldn't understand why other fine-tuning efforts failed until I see Figure 1.

Weakness / Suggestions:
The final analysis part is a bit weak (possibly due to the limitation of article length and the desire to squeeze things into 9 pages). I would suggest prioritizing Fig 2 and put Fig 3 (and more analyses like this) into the appendix.
The table 1 and 2 might require some caption to explain the 10%, 15%, 20% meaning.
best-performing results should be highlighted in Tables
Minor typo on page 4: "are to acquire acquire samples"


**Summary Of The Paper:**

The paper proposed a simple idea of using the snapshot ensembles instead of deep ensembles for active learning for uncertainty based methods. The paper compared their method with traditional DE and MCDO on three different dataset (CIFAR and Tiny ImageNet) with three differenty acquisition functions and showed that the SE + FT method achieved both a high accuracy as well as a low computation requirement.

**Summary Of The Review:**

I think this is a very interesting paper that made a simple idea that other people have tried actually working. The experiments are sound and the clarity is good, overall a good paper.

---

> ### Author Response · Authors · 2022-11-14
> **Thank you for your review**
>
> We appreciate your time in reviewing our paper and your constructive comments.
>
> ### Typos & Figures
>
> We appreciate your corrections. Other typos and potentially misleading words have been corrected in the revision. Please refer to the overall comment for the errata. Your suggestions make the paper look much more readable and easy to follow.
>
> ### More on Fine Tuning
>
> First of all, thank you for sharing your experience with fine-tuning experiments on active learning. When we first tried FT, the performance was very poor as reported in the [Rakesh and Jain, 2021] [https://arxiv.org/pdf/2104.00896.pdf] paper. However, as you emphasized as well, the most important thing for us is the performance of the **final episode**. In order to achieve better performance during the experiment, we tried to improve the performance of intermediate episodes (by increasing the sizes of the replay buffer, tuning the regularization, hyperparameter, etc.), but those attempts did not lead to a significant gain in the final episodes. Therefore, as we mentioned in the paper, what really matters is the performance of the final episode and the intermediate episodes are not important.

---

### Official Review · Reviewer_WKSq · 2022-10-25

**Confidence:** 4
**Correctness:** 2
**Technical Novelty And Significance:** 1
**Empirical Novelty And Significance:** 3
**Recommendation:** 3

**Clarity, Quality, Novelty And Reproducibility:**

The paper is overall clear and easy to follow. The major limitation is on the novelty and the comprehensiveness of the empirical results.

**Strength And Weaknesses:**

Strength

- The authors propose a simple modification of ensemble-based AL learning by using Snapshot ensemble and continual learning, which achieves comparable or better result compared to deep ensemble while reducing the computational cost, when used with simple uncertainty-based acquisition functions.
- The finding that snapshot ensemble produces more diverse predictions than deep ensemble is interesting.

Weakness

- The novelty of the work is limited, as both the snapshot ensemble and continual active learning have been introduced previously, and there is no novel acquisition function or AL algorithm.
- The authors only consider very basic AL methods (VR, Entropy, BALD) which are relatively old and has shown to be inferior than recent SOTAs. Moreover, the empirical experiments are incomplete, with full comparison among MC-dropout, DE, SE and continual-SE only available for VR, making the conclusion on superiority of SE unconvincing.
- The authors use a batched AL setting; however, the chosen acquisition functions are not suitable for batched AL as it just greedily takes the top m examples (in fact BALD can be worse than random in batched setting). It would be more compelling if result on algorithms designed for Batched AL is shown.
- The claim that prediction diversity is more important than model performance could be misleading, as overly diverse predictions could result in under-confidence, making choices of samples almost random. The right amount of uncertainty should be calibrated and informative of actual model uncertainty. No theoretical analysis is provided on why SE always produce better diversity.


**Summary Of The Paper:**

The authors propose to use snapshot ensemble (SE) to replace deep ensemble (DE) for modeling uncertainty in active learning. SE takes predictions from different epochs of training of a single model and thus is more efficient than DE. The authors compare SE with DE and MC-dropout on several basic AL acquisition functions including Variation Ratio, Max-entropy and BALD, and show that SE is comparable. They also use SE with continual learning where the model is not trained from scratch but is continuously finetuned after each acquisition. Such training strategy achieves comparative result with vanilla AL when using VR as acquisition.

**Summary Of The Review:**

The paper proposes a simple modification to ensemble-based AL by relacing DE with SE and shows some preliminary results using basic AL strategies. Although the result on VR is promising, more thorough comparison with recent AL techniques is needed to prove the significance of the result.

---

> ### Author Response · Authors · 2022-11-14
> **Thank you for your review**
>
> We appreciate your time in reviewing our paper and your constructive comments.
>
> ### Limited novelty of SE or FT
>
> While the snapshot ensemble has been popular in the literature, to the best of our knowledge, its application to active learning and demonstrating its superior performance even surpassing deep ensembles are novel. For continual active learning, as we pointed out in the general comment above, other continual AL papers are merely applied AL algorithms to continual learning settings with domain shifts, or it just points out that DE used in a continual learning setting shows bad performance, not proposing a fine-tuning based approach as ours.
>
> ### No acquisition function proposed
> As our title says, our goal is to propose a simple yet effective active learning strategy that can widely be applied to any uncertainty-based acquisition functions, not to propose a specific acquisition function. That is, since our main contribution is not specific to certain acquisition functions, in principle, it can be applied to any SOTA methods to further boost up performances (provided that the acquisition functions are uncertainty-based).
>
> ### Considering only basic acquisition functions and No SOTA acquisition functions
>
> Again, our goal in the paper is not to propose an elaborated acquisition function but to introduce an effective algorithm for measuring epistemic uncertainty from snapshot ensembles (+ fine-tuning based active learning approach).
>
> Many previous results in Active Learning (AL) literature have limited their scopes either to small-scale datasets (MNIST, FashionMNIST, etc.) or to datasets with high redundancy (CINIC-10, repeated MNIST). In many methods considered as SOTA, the experimental dataset is very limited. A recent study comparing reproduced results of SOTA acquisition functions shows poor performance of them on large-scale image datasets. [Zhan et al., 2022] [https://arxiv.org/abs/2203.13450] In particular, in the Tiny ImageNet results, the traditional acquisition functions (VR or ME) still show the best performance. Also, we compared our approach to the baselines under the setting that is made as general as possible, and ours consistently performed better than baselines (especially than DE which was reported to be the best) regardless of the choice of acquisition functions, datasets, and the way we train classifiers (single model or DE), which reassures our main contribution to present a simple and robust algorithm without sophisticated design for acquisition functions. Also, as Reviewer qYVA mentioned, the strength of this framework can easily be extended to various acquisition functions reported in the literature, especially the ones based on ensembles.
>
> ### The diversity-accuracy balance
>
> We agree on this point, and finding the right balance between diversity and accuracy would be an important problem to be tackled. However, please note that we never argued in our paper that we don’t care about the accuracy of predictions at all; if we did, there is no reason for us to train models in the first place.
>
> We also want to make clear that we have never said “SE always produces better diversity” in the paper. Instead, we rather emphasized the importance of acquiring diverse snapshots for AL such as the followings: “To obtain diverse parameters, the learning rate of the training run is carefully chosen to encourage the optimization path to explore a wide area of the loss surface (...)” (in Section 2.3); “The reason why SE is effective for AL is that it can build a committee of models yielding diverse predictions.” (at the beginning of Section 5); “We conjecture that SE builds a committee of models whose predictions are more diverse than the ones from other methods, and this is a key to its success in AL.” (in Section 5.2).
>
> We recommend collecting snapshots after a model is converged. If a learning rate during SE is too small, SE does not produce diverse predictions; if a learning rate is too large, then the learning trajectory becomes too unstable and eventually moves to meaningless areas. This result is quite obvious. We also included additional experiment results on different hyperparameter settings for SE in Table 5. Please refer to that as well as our response to another part of your comment: “Diversity may lead to under-calibrated results.”

---

> > ### Author Response · Authors · 2022-11-14
> > **Thank you for your review**
> >
> > ### Diversity may lead to under-calibrated results.
> >
> > **TL;DR.** We empirically showed that too much diversity may harm the acquisition quality for SE and proposed guidelines for selecting hyperparameters for SE. Still, diversity among predictors is important, especially with fewer training examples, and VR is robust to calibration (even when under-confident).
> >
> > For CIFAR-10 dataset, SE outperformed DE, and we conjecture that a high constant learning rate during the snapshot collection contributes to yielding diverse predictions. Each parameter snapshot has lower accuracy (about 10~20% lower) compared to the model before rewarming the learning rate. As shown in Fig. 2 and Fig. 3, SE models agree on correct examples with certainty while exhibiting diverse predictions on confusing ones. SE+FT also shows decent acquisition ability even with lower accuracy. This illustrates that diversity plays a crucial role, which has been underlooked in the AL literature so far.
> >
> > With additional experiments in response to your comment, we show the effectiveness of rewarming a learning rate after convergence. We included the results in Table 5. As you may have guessed, too high learning rates may cause a model to diverge and move to different mode values or meaningless areas, which surely degrades the acquisition quality. For example, some of the snapshots gathered with an SE learning rate of 0.05 or higher have shown a validation accuracy of 10% (the same as random guessing on CIFAR-10). The performance was not recovered even after the training trajectory was stabilized since it seemingly went too far from the optimal parameter space.
> >
> > For CIFAR-100, however, a high learning rate during SE causes an unstable learning trajectory, and rather a learning rate lower than the base learning rate was enough to collect diverse snapshots. Our additional findings regarding your comment can be summarized as follows.
> > * Diversity is still important, but too much degradation in performance results in less gain in acquisition quality.
> > * Nevertheless, VR, which is based on hard voting among predictors, shows robust performance compared to other acquisition functions which rely on calibration.
> > With these additional findings, we decided to come up with proposed guidelines for selecting hyperparameters for SE. We wouldn’t have done this if it were not for your review. Again, thank you for your hard efforts in reviewing our paper.
> >
> > ### Batch AL settings
> >
> > As you pointed out, the extension of our method to the batched AL setting is a natural future research direction. We actually tried BatchBALD [Batchbald: efficient and diverse batch acquisition for deep Bayesian active learning] but failed to obtain decent results on the datasets we tested, even with their original version of the algorithm (using MCDO for uncertainty estimation). We also tried our idea for BatchBALD (using SE for uncertainty estimation), but failed as well. We guess that other than relatively easy datasets (repeated MNIST or CINIC-10) originally tested in BatchBALD paper, it is not thoroughly validated for challenging real-world datasets.

---

### Author Response · Authors · 2022-11-14
**Errata and Overall Comments**

# Errata
### Overall
* Typos in Sections 1, 3, 4, 5

### Background
* Section 2.2: Replace “epistemic uncertainty” with “uncertainty”
* Section 2.2: Rewrite a sentence stating that BALD measures epistemic uncertainty.

### Methods
 * Section 3-1: “does not incur any additional cost” → “does not incur any additional computational cost for training” (clarification.)

### Experiments
* Section 5-1: Added additional results of ME+MCDO, SE+FT, MAR in CIFAR-10.
* Section 5-1: Added additional results of BALD, ME in CIFAR-100.
* Section 5-1: Replotted Fig 1 for visibility.
* Section 5-1: Effectiveness of fine-tuning paragraph – fixed the reference to Fig 1. due to LaTeX issue (pointed out by the reviewer s1VV: “I could not see where Figure 1 is referenced in the narrative.” and qtq1).
* Section 5-1: Added specific statement about baseline to “random baseline”
* Section 5-2: Reorganized sentences.
* Section 5-2. Figure 2. The histogram gets longer. The percentages now indicate the test error instead of the test accuracy to match VR scores for easier understanding.
* Section 5-2: Edited captions of Fig 2. and Fig 3.
* Section 5-2: “Fig. 2. (right) shows the ~”, “The percentages denote the test error for each bin, ~”, “Since unlabeled examples are labeled in the order of ~”, “For example, in the bin with VR=0.8 (all disagree), ~” Clarified the explanation of the plots.

### Appendix
* Re-organized the Appendix.
* Added a new section ‘Additional results and discussions for SE’ discussing hyperparameter configurations, and visualizations.
* Added additional experimental results.
* The number of data for fine-tuning → Replay buffer size.
* Added Figure 5. to show VR scores with DE is overconfident, and those with SE are much more reliable.

# Overall Comments
We thank all the reviewers for their constructive comments and feedback. They consider our approach to be simple and powerful (by WKSq, qtq1, qYVA, and s1VV) just like our title implies (by s1VV), somewhat novel (by s1VV and qtq1), and timely (by qYVA) in that it combines two ideas and successfully applies them in Active Learning (by WKSq, qtq1, and qYVA). Here, we address some common concerns raised by multiple reviewers.

### Novelty of fine-tuning based active learning
As pointed out by WKSq and qtq1, fast ensembling methods and continual learning techniques for AL have been used in academia as well as in practice for sure. To the best of our knowledge, the fine-tuning based active learning we propose is differentiated from the existing works related to continual AL, as summarized below:

* Continual Active Learning for Efficient Adaptation of Machine Learning Models to Changing Image Acquisition [Perkonigg et al., 2021] used continual learning in hypothetic domain shift settings due to different MRI imaging machines.
* Similarly, Continual Active Learning Using Pseudo-Domains for Limited Labelling Resources and Changing Acquisition Characteristics  [Perkonigg et al., 2021] tackled the same problem.
* Efficacy of Bayesian Neural Networks in Active Learning [Rakesh and Jain, 2021] showed that BNN outperformed DE in continual learning settings.
* Few-Shot Continual Active Learning by a Robot [Ayub and Fendley, 2022] – RL setting, robotics (continual learning is crucial)
* The Power of Ensembles for Active Learning in Image Classification [Beluch et al., 2018] evaluated implicit ensembling methods, but all implicit ensemble techniques showed lower accuracy compared to deep ensemble.

That is, existing works presenting “continual active learning” literally apply active learning frameworks to a continual learning setting where there are domain shifts in datasets; except for [Rakesh and Jain, 2021], none of them discusses a fine-tuning idea as we suggested - keeping a single model and continuously train it until the last episode. [Rakesh and Jain, 2021] discuss a fine-tuning-like idea, but they just show that the fine-tuning fails with their active learning algorithms. To summarize, ours is the **first fine-tuning based active learning algorithm that works well** (even surpassing deep ensembles) for real-world large-scale datasets.

### Additional results
In table 5, as suggested by the reviewer qYVA, we added results of the Margin uncertainty function well know to outperform other entropy uncertainty sampling. Also, in appendix A, we provide additional results about hyperparameters for SE and FT.
* Table 5 shows test accuracy on different hyperparameters like number of snapshots, SE learning rate and starting point of SE. The results of this table show that our method is not heavily dependent on hyperparameters.
* Table 7 provides test accuracy on different sizes of replay buffers for FT. The performance according to different sizes of replay buffers does not differ significantly in performance based on a specific number.

Also, we added Figure 5 shows that DE is overconfident relative to SE as the number of labeled data increases.

---

### Decision · Program_Chairs · 2023-01-20

**Decision:**

Accept: poster

**Justification For Why Not Higher Score:**

More theoretical results would have been needed to rate the paper higher.

**Justification For Why Not Lower Score:**

As the title suggests, the method is simple and useful, the claim made were verified experimentally.

**Metareview: Summary, Strengths And Weaknesses:**

The paper proposes a deep active learning method where the sampling is based on uncertainty estimates based on parameter "snapshots" from a single optimization path. The authors show that this method is effective, using a single learning trajectory for the active learning episodes as opposed to having to retrain the network. This method solves an important problem, relevant in the continual learning setting. Previous methods do not maintain a single model, use it to sample and successfully train it continuously, whereas previous attempts at this failed - the authors pointed this out in addressing reviewer WKSq's comments about novelty. The reviewer did not reply to the author's explanations of the differences to prior work, however, their explanations make sense to me. I urge the authors to make these differences to prior work clearer in the camera ready. Reviewer qtq1 pointed out that the analysis part in the main paper should be expanded. I fully agree with this, and their suggestions about moving the figures, because the method itself is straightforward (albeit effective), so gaining some insight into why it works is especially important. The authors have also addressed the questions of reviewer s1VV about hyper parameter tuning, learning rate and runtime, and performed additional experiments, as requested.
To summarize, this paper presents a simple method to perform active learning and train a deep model continuously, demonstrating its effectiveness on several datasets.

**Note From Pc:**

if the above contains the word "oral" or "spotlight" please see: "oral" presentation means -> notable-top-5% and "spotlight" means -> notable-top-25%. As stated in our emails, we are disassociating presentation type from AC recommendations

**Summary Of Ac-Reviewer Meeting:**

N/A